# Lean Finder: Semantic Search for Mathlib That Understands User Intents

**Jialin Lu**[1], **Kye Emond**[2], **Kaiyu Yang**[3] **Swarat Chaudhuri**[4], **Weiran Sun**[1], **Wuyang Chen**[1]

[1]Simon Fraser University    [2]University of Waterloo    [3]Independent Researcher
[4]University of Texas at Austin

## Abstract

We present **Lean Finder**, a semantic search engine for Lean and mathlib that understands and aligns with the intents of mathematicians. Progress in formal theorem proving is often hindered by the difficulty of locating relevant theorems and the steep learning curve of the Lean 4 language, making advancement slow and labor-intensive. Existing Lean search engines, though helpful, rely primarily on informalizations (natural language translation of the formal statements), while largely overlooking the mismatch with real-world user queries. In contrast, we propose a **user-centered semantic search** tailored to the needs of mathematicians. Our approach begins by analyzing and clustering the semantics of public Lean discussions, then fine-tuning text embeddings on synthesized queries that emulate user intents. We further align Lean Finder with mathematicians' preferences using diverse feedback signals, encoding it with a rich awareness of their goals from multiple perspectives. Evaluations on real-world queries, informalized statements, and proof states demonstrate that our Lean Finder achieves over 30% relative improvement compared to previous search engines and GPT-4o. In addition, Lean Finder is compatible with LLM-based theorem provers, bridging retrieval with formal reasoning. Lean Finder is available at: `https://leanfinder.github.io`.

## 1 Introduction

Advances in Lean and mathlib (De Moura et al., 2015; Moura & Ullrich, 2021) are turning mathematical discovery into a collaborative and verifiable research workflow. On this foundation, provers powered by large language models (LLMs) have sprinted ahead (AlphaProof & teams, 2024; Xin et al., 2024a;b; Ren et al., 2025; Lin et al., 2025a). Parallel progress in autoformalization has shifted from hand-written grammars to few-shot LLM translation, and large corpora generated through back-translating Lean code ("informalization") have bootstrapped even stronger translators (Wu et al., 2022; Jiang et al., 2024; Lu et al., 2024).

Despite these advances, state-of-the-art LLMs still cannot solve math research problems. This underscores the continued need for substantial **human efforts**, which unfortunately remain slow and labor-intensive for **two bottlenecks**. First, locating the right theorem is frustrating: search tools are rudimentary, naming conventions are often inconsistent (Zulip, 2021b; 2020b), and high-quality Lean examples remain scarce. Moreover, as we will demonstrate in our experiments (Table 2), identifying the correct theorem is challenging not only for humans, but also for state-of-the-art LLMs. Second, Lean's syntax, grammar, and tactics incur a steep learning curve. Even veteran mathematicians and experienced programmers regularly report difficulties when writing Lean (Zulip, 2021a; 2020a). To facilitate mathematicians, **semantic search is vital to Lean formalization and theorem proving**. Recent search engines take informal statements or live proof states and retrieve mathlib4 lemmas or tactic suggestions, thereby aiding human Lean users (Gao et al., 2024a;b; Tao et al., 2025a; Shen et al., 2025; Ju & Dong, 2025; Asher, 2025). However, these search engines target machine translation rather than human use. They "informalize" statements from a supposedly neutral viewpoint, whereas **real users bring inherent bias**, typically seeking explanations from their own specific perspective.

Consider the two queries below. The first is an informalization of a formal statement that current Lean search engines handle (Gao et al., 2024a;b; Ju & Dong, 2025; Asher, 2025):

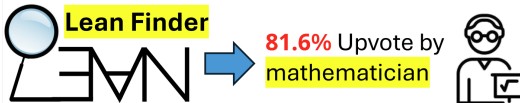

Figure 1: In the evaluation with user queries, real users preferred Lean Finder in 81.6% of cases, compared with 56.9% for Lean Search Gao et al. (2024a) and 54.1% for GPT-4o.

**Query 1:** Denote $L/K$ a field extension, $x, y$ in $L$ are algebraic elements over $K$ with the same minimal polynomial. Then the $K$**-algebra isomorphism** $algEquiv$ **between the simple field extensions** $K(x)$ **and** $K(y)$ **maps** the generator $x$ of $K(x)$ to the generator $y$ of $K(y)$, i.e., $algEquiv(x) = y$.

**Target Statement 1:**

```
1  theorem algEquiv_apply {x y : L} (hx : IsAlgebraic K x) (h_mp :
       minpoly K x = minpoly K y) : algEquiv hx h_mp (AdjoinSimple.gen K
       x) = AdjoinSimple.gen K y := -- proof omitted for brevity
```

However, a human query[1] in practice may look like:

**Query 2:** I'm working with algebraic elements over a field extension and I have two elements, say $x$ and $y$ in $L$. I know $x$ is algebraic over $K$, and I've shown that y is a root of the minimal polynomial of $x$. **Does this imply that the minimal polynomials of** $x$ **and y are actually equal**?

**Target Statement 2:**

```
1  theorem eq_of_root {x y : L} (hx : IsAlgebraic K x) (h_ev :
       Polynomial.aeval y (minpoly K x) = 0) : minpoly K y = minpoly K
       x) := -- proof omitted for brevity
```

Although both queries involve a similar mathematical context (algebraic elements $x, y$ in a field extension $L/K$ and their minimal polynomials), **they have different concerns:** the first asserts an isomorphism between field extensions (generated by two algebraic elements with the same minimal polynomial), whereas the second seeks to decide whether the two minimal polynomials are equal from the fact that $y$ is a root of $x$'s minimal polynomial. This user *latent* (motivation, perspective, abstraction) cannot be inferred or encoded by a purely syntactic informalization. Addressing this challenge calls for Lean search engines that can understand a mathematician's intent, not merely surface-level matches. We defer a more rigorous analysis in Section 2.2, and ask our core question:

> *How to make the Lean search engine user-centered and tailored to understand the intents and needs of working mathematicians?*

In this paper, we aim to build a search engine for Lean and mathlib4 that captures the diverse intents of mathematicians. Our approach **analyzes and clusters public discussions**, then **synthesizes queries that simulate user intents** (Section 3.1). We incorporate Lean code from a wide variety of sources, and support multiple input types such as informalizations and proof states (Section 3.2). With the diverse data, we fine-tune LLM embeddings to achieve both strong retrieval performance and alignment with user preferences (Section 3.3). Lean Finder is extensively evaluated on different input types and real user queries (Section 4.1, Section 4.2). It is released as a web service (Figure 8) to support mathematicians coding in Lean. Finally, Lean Finder is compatible with LLM-based theorem provers via **prover-agnostic retrievals**, serving as a plug-and-play tool to boost prover performance without retraining (Section 4.3).

We summarize our contributions below:

1. When evaluating on real-world queries by real users, our Lean Finder is upvoted 81.6% compared to 56.9% from Lean Search (Gao et al., 2025) and 54.1% from GPT-4o in an LMArena-style test (Chiang et al., 2024).

2. Lean Finder outperforms GPT-4o and other Lean search engines, achieving 30%+ and 16%+ relative gains (Recall@1) when queried with informalized statements and proof states, respectively.

---

[1]We synthesized this "Query 2" (see Section 3.1), as we do not release any real user queries due to privacy concerns (Section N).

3. Lean Finder can also boost the performance of LLM-based provers via in-context retrieval augmentation, making it a prover-agnostic retrieval model.

4. We release the largest code search dataset for Lean repositories, consisting of over 1.4M query-code pairs (582,102 synthesized user queries, 244,521 informalized statements, 337,647 proof states, and 244,521 formal statements).

## 2 MOTIVATION

### 2.1 BACKGROUND: SEARCH ENGINE FOR LEAN AND MATHLIB

Lean (De Moura et al., 2015; Moura & Ullrich, 2021) is an interactive theorem prover based on dependent type theory, with a community-run library (mathlib4) containing over 230k theorems and 110k definitions. This offers far more "entry points" than typical programming platforms (Python exposes 89 built-in functions; C++ standard library has 105 headers). **Finding the right lemma in this vast library is difficult**: existing tools like #find, library_search, and Loogle[2] depend on exact names or goal states and often fail when naming conventions drift or examples are sparse. Recent LLM-based search engines (Gao et al., 2024a;b; Yang et al., 2023; Tao et al., 2025a; Shen et al., 2025; Ju & Dong, 2025; Asher, 2025; Zhu et al., 2025) embed informal queries or proof goals, but remain optimized for embedding similarity rather than the nuanced, shifting intents of mathematicians. In short, sheer library scale and rudimentary search together make locating lemmas a first-order pain point, even before one tackles Lean's non-trivial syntax and tactic language.

### 2.2 CURRENT SEARCH ENGINES MISALIGNED WITH MATHEMATICIANS' NEEDS

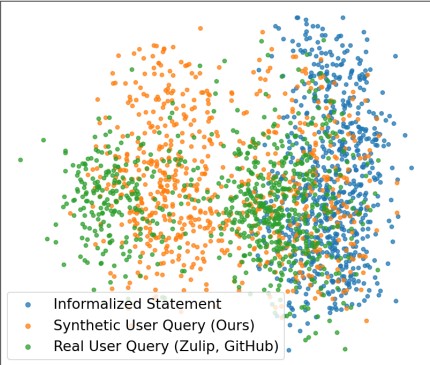

Figure 2: PCA of Lean/mathlib-related queries from different sources. Our synthesized user query (orange, Section 3.1) aligns more closely with real queries (green), whereas direct informalization (Gao et al., 2024b) (blue) shows distributional deviations. Real user queries are from Zulip and GitHub.

Despite notable progress, current Lean search engines are optimized for informalized statements, instead of questions mathematicians actually pose. Most systems embed informal descriptions (natural languages) of statements or proof states, together with possible context or dependencies, then retrieve matching lemmas (Yang et al., 2023; Gao et al., 2024a;b; Tao et al., 2025a; Shen et al., 2025). Because these informalizations are generated by LLMs rather than user logs, we have no guarantee that their wording, granularity, and purpose align with the queries real Lean practitioners type. **This unexamined domain gap between LLMs' informalizations and authentic human queries has so far been largely ignored**, leaving retrieval systems potentially over-tuned to deviated distributions.

To justify this domain gap, we use PCA to visualize distributions of different query embeddings generated by all-MiniLM-L6-v2 (Wang et al., 2020). As shown in Figure 2, the cluster of LLM-generated informalizations (blue) only spans over a subspace of the cluster by queries collected from Zulip and Github (green), highlighting a clear domain gap between real user queries and surrogate ones on which current Lean search engines are trained. By contrast, embeddings of our synthesized user queries in orange (Section 3.1) are much more aligned with the human cluster, suggesting that they better capture the practical needs of mathematicians coding in Lean.

---

[2]https://loogle.lean-lang.org/

# 3 LEAN FINDER: DATA SYNTHESIS, CODE SEARCH, HUMAN ALIGNMENT

## 3.1 USER-CENTERED QUERY SYNTHESIS

Section 2.2 underscores that effective Lean retrieval for mathematicians requires authentic user queries, yet assembling such data at scale is obstructed by two bottlenecks:

1. The volume of publicly available Lean questions by real users is tiny (for example, we are only able to fetch 693 answerable user queries from Zulip/GitHub), which is orders of magnitude smaller than the billions-token corpora consumed by modern LLMs;

2. Even with real user queries, tracing the precise answer (formal statement) that eventually resolves each query is infeasible, due to open math problems and evolving Lean/mathlib4 development.

Therefore, preparing a large fully annotated set of genuine user queries is unrealistic. Instead, we propose a novel reverse strategy to synthesize a large amount of diverse and realistic queries, as overviewed in Figure 3. The **core idea** is: based on different perspectives that mathematicians ask Lean/mathlib4-related questions (Section 3.1.1), we prompt LLM generation to simulate mathematicians' intents (Section 3.1.2).

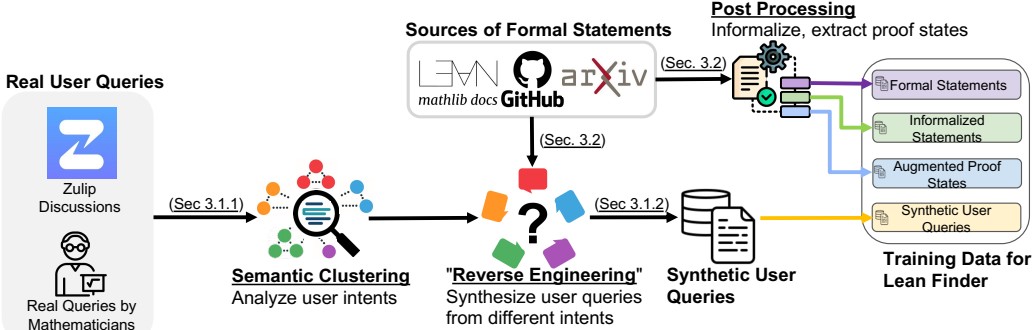

Figure 3: We synthesize user queries as part of our training data. First, we analyze the semantics of public Lean/mathlib queries posted by real mathematicians and cluster them into user intents (Section 3.1.1). These intents then prompt the generation of synthetic user queries from formal Lean statements collected from mathlib, GitHub, and research papers (Section 3.1.2). The synthesized user queries form the core part of our training data for our Lean Finder. To generalize Lean Finder across diverse inputs, we further include additional modalities (formal/informal statements and proof states, see Sections 3.2) extracted from formal Lean statements.

### 3.1.1 SEMANTIC ANALYSIS OF REAL HUMAN QUERY

The first step for user query synthesis is to systematically collect real user discussions, and then analyze the diverse semantics and intents of mathematicians coding in Lean.

**Collect User Discussions.** To achieve this semantic analysis, we utilize Lean Zulip Chat[3], the primary discussion forum for Lean, as well as GitHub, as the main sources of real-world queries. We use the Zulip API to extract high-quality discussions from five active and thematically rich channels (*new members*, *lean4*, *mathlib4*, *Is there code for X*, *metaprogramming/tactics*), retaining up to the first five user messages, which typically capture the core query. We then prompt GPT-4o (Appendix O.2) to filter for questions answerable by Lean statements and paraphrase the main query articulated in the messages. This process yields 693 real user discussions from Zulip and Github[4].

**Cluster User Intents.** Mathematicians may ask Lean/mathlib4-related questions from different perspectives (see examples in Section 1 and our analysis in Figure 2), therefore we need to identify the distinct types of queries collected from Zulip/Github. We first prompt OpenAI o3 to bootstrap the initial set of clusters from a subset of collected queries. Then, the same model is fed with more user discussions and is prompted to iteratively update existing clusters or introduce new clusters if necessary.

---

[3]https://leanprover.zulipchat.com/

[4]Due to privacy concerns, we do not release either Zulip or GitHub collections.

Table 1: Five clusters of mathematicians' intents categorized from our collections of real user queries from Zulip and GitHub.

| Intent Cluster | Semantic Definition | Examples |
|---|---|---|
| Searching for existing code / lemmas | Whether a definition, data-structure implementation, lemma, or theorem is already available in Mathlib/Lean, or if there is an easy way to get the desired statement. | "Is there code for ...", "Do we have ...", "Is there a lemma which ...", "Where can I find ..." |
| Meta- / tactic- programming questions | Questions about Lean 4 metaprogramming: writing tactics or elaborators, creating macros, manipulating Expr or environments, controlling metavariables, interacting with 'simp'/'aesop'/'omega' from meta code. | "Why doesn't the code give an error about a cycle existing when using Lean.MVarId.assign?", "Is there a way to make my first approach work to get Exp out of Q(Exp) using expr% macro?" |
| Type-class, instance, axiom | Proofs fail, or definitions cannot be written because Lean cannot find (or should not use) certain type-class instances, or users need advice on constructing/avoiding such instances and on the correct logical meaning of such instances. | "How to define or derive instances...", "Why certain instance-search problems happen...", "Whether particular instances should exist at all..." |
| Proof engineering & everyday Lean usage | Concrete goals or failing scripts around practical, day-to-day proof writing in Lean. | How to: finish or shorten proofs, make simp, rw, omega, linarith, etc. succeed, rewrite or unfold expressions, handle coercions and subtypes, manipulate logical connectives ($\forall, \exists, \wedge, \vee$), pattern-match over inductives, or split cases. |
| Library design & large-scale formalization | Conversations that concern the construction of large-scale statements: proposing new mathematical structures or tactics, refactoring existing ones, performance trade-offs, big formalizations or theorems. | "How can I apply the coYoneda lemma or density theorem to simplicial sets valued in an arbitrary universe without restricting the variable?" |

See Appendix O.3 and Appendix O.4 for the prompts. This process results in five distinct and semantically meaningful clusters of queries, as shown in Table 1, capturing a comprehensive range of intents from which mathematicians ask about Lean/mathlib4.

### 3.1.2 Query Synthesis with User Intents

Based on semantic clusters of real queries in Table 1, we synthesize a large-scale dataset of queries simulating diverse user intents. Although it is typically challenging to resolve a query with precise formal statements in Lean, we can still **reversely synthesize the query, instead of the ground truth**. Specifically, we assume each formal Lean/mathlib4 statement could be the true answer to some unknown user queries, and now the key is to simulate queries with realistic user intents.

We aim to instruct the LLM to generate plausible user queries that align with the specified cluster's perspective while remaining grounded in the given formal content. To avoid forcing every statement into all intent clusters, we adopt a two-step procedure. *First*, we prompt GPT-4o to analyze the given Lean statement and determine which query clusters from Table 1 are applicable. This filtering step ensures that only semantically meaningful queries are generated, avoiding unnatural cases where a cluster is irrelevant to a statement. *Second*, for each selected cluster, we prompt GPT-4o with rich context information (e.g. formal/informal Lean statement, cluster information) to synthesize the queries. See Appendix O.5 for our prompt. This procedure yields a total of 582102 synthetic queries distributed across five distinct intent clusters, enabling fine-tuning of embedding models to better meet mathematicians' needs on Lean search.

We visualize the distribution of clusters in synthetic user queries in Figure 4 (a).

### 3.2 The Largest Code Search Dataset for Lean Repositories

Beyond the synthesized user queries discussed above, we additionally incorporate the following data sources and modalities, releasing the largest **code search** dataset for Lean and mathlib4.

**Research-driven Data Collection.** While mathlib4 is the main source of formal statements for Lean, it does not capture the full breadth of formal mathematics. To address this, we expand beyond mathlib4 and include GitHub repositories linked to recent research papers and domain-specific

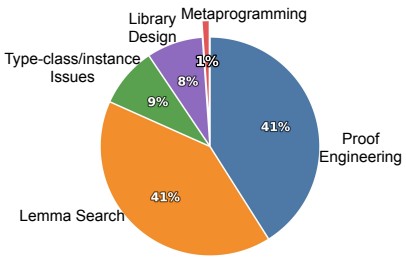 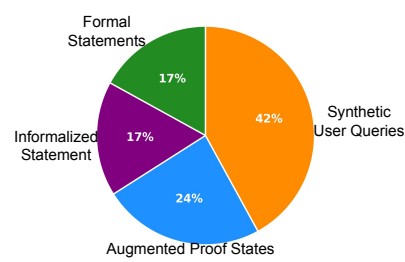

(a) Distribution of clusters in synthetic user queries.     (b) Distribution of input modalities.

Figure 4: Distribution of our training dataset. (a) At a fine granularity, our synthetic user queries distribute across five clusters of user intents (identified in Section 3.1.1). This distribution also aligns with the weekly popularity trends of relevant channels on Lean Zulip Chat (Appendix B.5). For detailed descriptions and examples of these clusters, see Table 1. (b) Overall, our dataset contains a total of 1.4M pairs of queries and formal Lean statements, distributed across four input modalities.

libraries. This ensures our search engine is trained on diverse, up-to-date statements that reflect both everyday Lean practice and cutting-edge math research. See Appendix B.1 for the detailed sources of our formal Lean statements.

**Informalization.** To support retrievals by both informalized statements and the generation of synthetic user queries (Section 3.1.2), we convert formal Lean statements into natural language descriptions. Inspired by Herald (Gao et al., 2024b), for each Lean statement, we provide rich context information for GPT-4o to better understand the Lean statement and provide a good quality informalization. The context includes information like dependent statements, related statements, neighbor statements, along with their informalization. See Appendix B.3 for details about informalizations.

**Other Input Modalities.** To better capture user intent, we introduce augmented Lean proof states, which enrich a Lean proof state (subgoals with context and target) by describing the intended direction of progress. These descriptions are synthetically generated from Lean proofs with GPT-4o, providing natural explanations of proof transitions that mirror how users query the system when searching for applicable lemmas, enabling more fine-grained retrieval (see Appendix B.4 for details). We also include formal statements, which retain only the declaration without the proof body, supporting cases where users recall a statement only partially or seek semantically similar ones. Lean Finder can thus retrieve relevant statements even from imprecise inputs. We visualize the distribution of input modalities in Figure 4 (b). More statistics about our training data are summarized in Appendix B.2.

## 3.3 TRAINING LEAN FINDER WITH CODE SEARCH

We train our code search model, **Lean Finder**, with a two-stage pipeline (Figure 5). In the first stage, we employ a contrastive learning objective to establish a general alignment between queries of diverse input modalities and formal Lean statements (Section 3.3.1). In the second stage, we incorporate user votes collected via our web service, along with auxiliary feedback from LLMs, to further refine Lean Finder, aligning retrieval more closely with mathematicians' actual preferences (Section 3.3.2).

### 3.3.1 CONTRASTIVE LEARNING FOR LEAN CODE SEARCH

We adopt DeepSeek-Prover-V1.5-RL 7B (Xin et al., 2024b) as the base model for fine-tuning, for its extensive training on Lean 4 syntax and theorem proving tasks.

As a decoder-only architecture, only the final token in each sequence has access to the full context due to the causal self-attention. Therefore, we extract the final hidden state of the last token in the last decoder layer to embed the entire sequence. The model is fine-tuned with contrastive loss on "informal query $q_i$ – formal code $c_j$" pairs, which aims to align the embeddings of matching pairs in a shared embedding space.

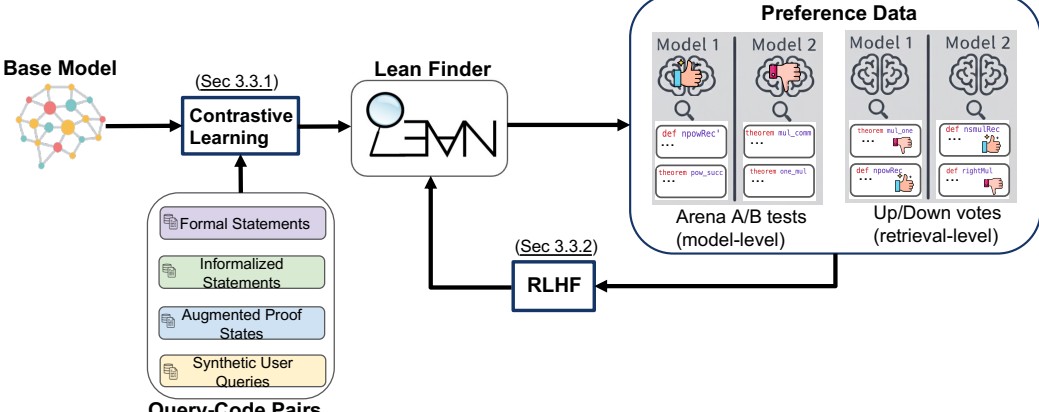

Figure 5: Lean Finder training pipeline. We first fine-tune the base LLM with contrastive learning (Wang et al., 2023) on four input modalities in our training dataset (Figure 4, synthetic queries, informalized statements, augmented proof states, and formal statements) to build a strong retrieval model for Lean (Section 3.3.1). We then deploy our Lean Finder as a web service and collect real users' retrieval preferences, which serve as human feedback. These preferences are used to further improve Lean Finder via Direct Preference Optimization (DPO (Rafailov et al., 2024), Section 3.3.2), resulting in a human-aligned boosted Lean Finder.

**Training data and in-batch negatives.** We construct sample groups of size $G$. For each query $q_i$, we pair one ground-truth Lean statement $c_i^+$ and attach $G-1$ negatives $\{c_{i,k}^-\}_{k=1}^{G-1}$ drawn from the statement corpus. With batch size $B$, a training batch contains $B$ such groups, yielding the candidate code set:

$$\mathcal{C}_b \;=\; \{\,c_i^+\,\}_{i=1}^B \;\cup\; \bigcup_{i=1}^B \{\,c_{i,k}^-\,\}_{k=1}^{G-1}, \quad \text{with} \quad |\mathcal{C}_b| = B \cdot G.$$

For a given query $q_i$, we apply token-level augmentation to obtain $\tilde{q}_i$, enhancing robustness to noisy or partial queries (e.g., misspellings or incomplete recollection), and we treat *all* codes in $\mathcal{C}_b$ except its paired positive $c_i^+$ as negatives; thus the number of negatives per query is $B \cdot G - 1$, combining intra-group negatives and code from all other groups (in-batch negatives).

**Contrastive Loss.** Let $\tau$ denote a temperature. We optimize the contrastive loss over in-batch candidates, where $\mathrm{Sim}(\cdot, \cdot)$ is the cosine similarity between two embeddings:

$$\mathcal{L}_{\mathrm{contrastive}} \;=\; -\frac{1}{B} \sum_{i=1}^B \log \frac{\exp\big(\mathrm{Sim}(\tilde{q}_i, c_i^+)/\tau\big)}{\sum_{c \in \mathcal{C}_b} \exp\big(\mathrm{Sim}(\tilde{q}_i, c)/\tau\big)}.$$

### 3.3.2 PREFERENCE ALIGNMENT

After initial training, Lean Finder shows strong retrieval performance across input modalities. To further adapt it to real user needs, we deployed it as a web service where users can provide feedback on retrieval results. We align the retriever with these preferences using Direct Preference Optimization (DPO) (Rafailov et al., 2024).

**Preference data.** Our preference dataset $\mathcal{P} = \{(q_i, c_i^+, c_i^-)\}_{i=1}^N$ consists of triplets where $q$ is a user query and $c^+$, $c^-$ denote a chosen and rejected Lean statement, respectively. It is built from a unique combination of user and LLM feedback. We collect preferences at two hierarchical levels. *First*, individual retrievals are explicitly upvoted or downvoted by users on our web service. *Second*, blinded model-wise preferences between Lean Finder and Lean Search (Gao et al., 2024a) are obtained both from users' online selections (see Figure 9) and from retrievals of real community queries on Lean Zulip, judged by GPT-4o. This design blends direct human feedback with LLM-based evaluations, and yields a high-quality and heterogeneous resource of 1154 triplets for aligning with user preferences. See Appendix E for more details.

**DPO fine-tuning.** We adapt the original DPO objective (Rafailov et al., 2024) to retrieval, by replacing sequence likelihoods with probabilities defined over candidate statements, computed from query–code similarity scores. Let $\theta$ denote the current retriever (policy) and $\theta_{\mathrm{ref}}$ denote the reference

retriever obtained from initial contrastive training. The adapted DPO objective for retrieval is simplified to

$$\mathcal{L}_{\text{DPO}} = -\mathbb{E}_{(q,c^+,c^-)\sim\mathcal{P}} \left[ \log \sigma \Big( \beta \big( \big( \text{Sim}_\theta(q,c^+) - \text{Sim}_\theta(q,c^-) \big) - \big( \text{Sim}_{\theta_{\text{ref}}}(q,c^+) - \text{Sim}_{\theta_{\text{ref}}}(q,c^-) \big) \big) \Big) \right],$$

where $\sigma$ is the sigmoid function and $\beta$ controls the deviation of the model from the reference. To prevent large degradation of general retrieval performance, we jointly train with the contrastive loss $\mathcal{L}_{\text{contrastive}}$ defined in Section 3.3.1. Let $\lambda$ represent the weight given to contrastive loss, the fine-tuning objective for this stage becomes

$$\mathcal{L} = \mathcal{L}_{\text{DPO}} \ + \ \lambda \, \mathcal{L}_{\text{contrastive}}.$$

## 4 EXPERIMENTS

Our data collection pipeline involves the OpenAI API, and our configurations are shown in Appendix C. Our training settings are in Appendix D.

Our experiments primarily compare Lean Finder with other Lean search engines across different input modalities, with particular emphasis on user-centered queries reflecting mathematicians' intents. In Section 4.1, we evaluate retrieval performance across input modalities, followed by a user study on real queries in Section 4.2. Section 4.3 explores Lean Finder as a retrieval backbone for LLM-based provers, and Appendix K reports ablation studies on the impact of preference alignment and using synthetic queries in training. Appendix J reports comparisons with additional baselines such as Lean Explore (Asher, 2025) and Qwen3-Embedding-8B (Zhang et al., 2025).

### 4.1 LEAN FINDER DEMONSTRATES STRONG RETRIEVAL PERFORMANCE

To evaluate the performance of our Lean Finder across different inputs, we compare it against recent search engines that support corresponding modalities. For the input modalities of *informalized statement*, *synthetic user query*, and *formal statement*, we compare with Lean Search[5] (Gao et al., 2024a;b) and GPT-4o. For *proof state input*, we compare with Lean State Search[6] (Tao et al., 2025b), Real Prover Search (Shen et al., 2025), and GPT-4o.

**Metrics.** We assess retrieval performance using the following metrics, both are higher the better:

- **Recall@K**: This measures the proportion of relevant results within the top K retrieved results.
- **Mean Reciprocal Rank (MRR)**: Computes the mean of the reciprocals of the first relevant result's rank across $N$ queries: $\text{MRR} = \frac{1}{N}\sum_{i=1}^{N}\frac{1}{\text{rank}_i}$. MRR is omitted for GPT-4o because it often generates fewer than 10 candidates.

**Settings.**

- To enable a **fair comparison** with Lean Search (Gao et al., 2024a;b), which relies on a fixed database of Lean statements, we require that every testing sample appears in its database.
- Since GPT-4o is not a retrieval model, we enable its web search tool and restrict its outputs to Lean statement names and evaluate with two criteria: 1) full-name match, which requires the exact full identifier of the Lean statement to match the ground truth; 2) stem match, a looser criterion where only the final component of the identifier must match (e.g., `DirectSum.IsInternal.coeAlgEquiv` counts as correct if the ground truth ends with `coeAlgEquiv`).
- Our test set includes 1000 informalized statements, 1000 formal statements, 1000 synthetic user queries, and 2224 proof states. We augment the formal statement inputs by replacing 20% of the query tokens with random tokens from vocabulary to simulate noisy queries. For proof states, we evaluate both the augmented version (with natural language guidance, Section 3.2) and the raw version (without augmentation) against other search engines. See Appendix G for detailed settings.

---

[5]https://leansearch.net/
[6]https://premise-search.com/

**Results.** Table 2 and Table 3 report retrieval performance across input modalities. Lean Finder consistently outperforms all baselines. Notably, GPT-4o with web search performs poorly even under the relaxed stem-match strategy, which can inflate scores. This suggests that while the latest LLMs may demonstrate strong generation performance, they struggle to accurately retrieve dependent statements, highlighting the need for specialized systems such as Lean Finder. While Lean Search and Lean State Search target informalized statements and raw proof states, Lean Finder surpasses them as well, including on raw proof states that it was never explicitly trained on. These results highlight Lean Finder's ability to generalize across diverse query formats and capture user intents, making it an effective tool for supporting real-world Lean theorem proving workflows.

Table 2: Comparison of Lean Finder with Lean Search and GPT-4o on retrieval performance across three input modalities: informalized statement, synthetic user query, and augmented statement. MRR is omitted for GPT-4o because it often generates fewer than 10 candidates. The best results are presented in **bold**.

| Model | Informalized Statement | | | | Synthetic User Query (Sec. 3.1) | | | | Augmented Statement | | | |
|---|---|---|---|---|---|---|---|---|---|---|---|---|
| | R@1 | R@5 | R@10 | MRR | R@1 | R@5 | R@10 | MRR | R@1 | R@5 | R@10 | MRR |
| Lean Finder | **64.2** | **88.9** | **93.3** | **0.75** | **54.4** | **84.4** | **91.4** | **0.68** | **82.7** | **97.0** | **97.7** | **0.89** |
| GPT-4o (full name match) | 14.8 | 20.9 | 22.6 | — | 13.6 | 21.5 | 23.3 | — | 39.7 | 42.9 | 44.1 | — |
| GPT-4o (stem match) | 21.1 | 28.4 | 30.0 | — | 17.8 | 27.4 | 30.5 | — | 48.2 | 52.6 | 54.1 | — |
| Lean Search | 49.2 | 76.5 | 82.5 | 0.61 | 47.1 | 77.7 | 83.7 | 0.60 | 59.2 | 81.9 | 85.5 | 0.69 |

Table 3: Comparison of Lean Finder with Lean State Search, Real Prover Search, and GPT-4o on retrieval performance across two input modalities: augmented proof state and raw proof state. "Augmented proof state": proof state augmented with natural language description of proofs (Section 3.2). "Raw proof state": proof state without the augmentation. MRR is omitted for GPT-4o because it often generates fewer than 10 candidates. The best results are presented in **bold**.

| Model | Augmented Proof State | | | | Raw Proof State | | | |
|---|---|---|---|---|---|---|---|---|
| | R@1 | R@5 | R@10 | MRR | R@1 | R@5 | R@10 | MRR |
| Lean Finder | **24.6** | **56.8** | **67.9** | **0.40** | **8.3** | **30.1** | **40.0** | **0.19** |
| GPT-4o (full name match) | 7.4 | 13.3 | 15.0 | — | 4.5 | 8.5 | 9.9 | — |
| GPT-4o (stem match) | 10.1 | 17.3 | 19.7 | — | 6.4 | 11.5 | 13.6 | — |
| Lean State Search | 4.99 | 27.7 | 39.6 | 0.16 | 3.3 | 23.1 | 32.1 | 0.13 |
| Real Prover Search | 8.0 | 29.0 | 39.2 | 0.18 | 7.1 | 26.2 | 34.3 | 0.16 |

## 4.2 LEAN FINDER IS PREFERRED BY REAL LEAN USERS

To evaluate Lean Finder on real-world queries, we conduct a user study with 5 participants on 128 GitHub-sourced queries. For each query, participants examine three retrievals from each model and rank them according to how likely they are to resolve the query (ties allowed). Model identities are hidden and their orders are shuffled. Participants can also choose "All Bad" or "I don't Know" to indicate that all models' responses are bad or they don't know how to answer. We record each model's frequency of being ranked at 1st/2nd/3rd place and its percentage of top-3 appearances among valid ballots (i.e., queries not marked as All Bad/I don't know). We also compute the normalized Borda score as $\text{Borda}_{\text{norm}}(m) = \frac{3 \cdot n_1(m) + 2 \cdot n_2(m) + 1 \cdot n_3(m)}{3N}$, where $n_1(m), n_2(m), n_3(m)$ are the counts of 1st/2nd/3rd place votes for model $m$ and $N$ is the number of valid ballots. Further details are provided in Appendix F.

The user study result in Table 4 shows that Lean Finder is strongly preferred over baselines. It receives the most best votes (139), almost twice as many as baselines. It also achieves a much higher Top–3 rate of 81.6%, compared to 56.9% for Lean Search and 54.1% for GPT-4o. Its normalized Borda score (0.67) is also much higher than baselines, indicating that participants consistently rank Lean Finder's results as more helpful. These results confirm that Lean Finder better aligns with user preferences and provides more helpful retrieval results in real-world query settings.

## 4.3 LEAN FINDER IS COMPATIBLE WITH LLM PROVERS

To examine whether Lean Finder can act as a prover-agnostic retrieval tool, we integrate Lean Finder with whole-proof generation models like Goedel Prover (Lin et al., 2025a) and DeepSeek-Prover V1.5 (Xin et al., 2024b), as well as step-wise provers like REALProver (Shen et al., 2025), following

Table 4: User ranking outcomes over 3 retrieval models, where $n_1(m)$, $n_2(m)$, $n_3(m)$ refers to the counts of 1st/2nd/3rd place votes for model $m$. Best results are in **bold**.

| Model ($m$) | $n_1(m)$ | $n_2(m)$ | $n_3(m)$ | Top–3 (%) | Borda$_{norm}$ |
|---|---|---|---|---|---|
| Lean Finder | **139** | **56** | 36 | **81.6**% | **0.67** |
| Lean Search | 70 | 51 | **40** | 56.9% | 0.41 |
| GPT–4o | 71 | 46 | 36 | 54.1% | 0.40 |

a retrieval-augmented generation (RAG) setup where Lean Finder provides candidate lemmas during proof attempts. Evaluation on MiniF2F (Zheng et al., 2022), ProofNet (Azerbayev et al., 2023), PutnamBench (Tsoukalas et al., 2024), and FATE-M (Shen et al., 2025) shows that integrating Lean Finder yields performance that is mostly consistent with the non-integrated setting, with modest improvements on certain benchmarks. The detailed results and discussion can be found in Appendix I.

## 5 RELATED WORKS

Prior efforts in code search (Husain et al., 2019; Wang et al., 2023), Lean retrieval (Gao et al., 2024a;b; Asher, 2025; Lean FRO; Morph Labs), and autoformalization (Jiang et al., 2023; Lin et al., 2024; Wang et al., 2024; Ying et al., 2024; Soroco et al., 2025) have advanced retrieval and translation between natural language and formal mathematics, but remain limited by narrow input types, paraphrase-level alignment, or brittle translations. Our work complements these directions by introducing a user-centered, intent-aware retrieval framework that explicitly models mathematicians' real query styles and integrates them into training, enabling Lean Finder to generalize across diverse modalities and substantially outperform existing systems on real user queries. Extended related work is provided in Appendix A.

## 6 CONCLUSION

We present Lean Finder, a significant advancement in the development of search tools for formal mathematics. By centering on real user intent and leveraging synthesized queries, semantic clustering, and fine-tuned embeddings, it bridges the gap between mathematicians' informal reasoning and formal theorem proving in Lean. Its superior performance over existing search tool demonstrate both practical utility and academic impact. Lean Finder not only enhances accessibility to mathlib4 but also lays the groundwork for future intelligent systems that assist in rigorous, collaborative mathematical discovery.

## THE USE OF LARGE LANGUAGE MODELS (LLMS)

LLMs did not play a significant role in either the research ideation or the writing of this paper. Their use was limited to correcting minor grammatical issues and typographical errors.

## ACKNOWLEDGMENT

This research used resources of the National Energy Research Scientific Computing Center, a DOE Office of Science User Facility supported by the Office of Science of the U.S. Department of Energy under Contract No. DE-AC02-05CH11231 using NERSC award NERSC DDR-ERCAP0034682 and ASCR-ERCAP0031463.

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

## A    EXTENDED RELATED WORKS

**Code Search and Lean Retrieval.**    Code search, aimed at retrieving the most semantically relevant code snippets from a codebase according to a specified natural language query, is a common activity that plays an important role in software development (Husain et al., 2019; Wang et al., 2023) and aligns closely with premise selection: both problems require retrieving a small set of code blocks that are helpful for downstream tasks. Early methods relied on lexical heuristics (Alama et al., 2014; Urban, 2004), while recent work leverages neural encoders (Irving et al., 2016; Wang & Deng, 2020; Mikuła et al., 2023; Yeh et al., 2023) and dense retrieval models (Karpukhin et al., 2020; Qu et al., 2020) by learning semantic embeddings. In formal domains like Lean, retrieval systems like Lean Search (Gao et al., 2024a;b) assist users by allowing them to search for relevant mathlib statements using natural language queries. These systems primarily function by matching the natural language translation of a formal statement to other formal statements in mathlib. While effective in some settings, this approach supports only narrow input types and treats informal queries as approximate paraphrases of formal code and focuses on surface-level alignment, without capturing the diverse intents behind real-world user questions or the broader context in which they arise. Lean Finder addresses this gap by introducing a user-centered, intent-aware retrieval framework for Lean that supports a broad range of input modalities and explicitly bridges the gap between the natural queries posed by mathematicians and the formal statements stored in Lean projects. By modeling real user intent and query diversity, Lean Finder achieves consistently strong retrieval performance across all modalities and demonstrates substantial gains over existing systems in a user study based on real Lean queries.

**AutoFormalization.**    Autoformalization, the automated translation of informal mathematics into formal proof-assistant code, has become a key strategy for addressing the scarcity of high-quality formal data. Early neural encoders for statement formalization (Wang et al., 2018) have been eclipsed by LLM-based pipelines that translate at scale with few-shot prompting or fine-tuning (Wu et al., 2022; 2024; Xin et al., 2024a;b; Soroco et al., 2025). Systems such as DeepSeek-Prover, InternLM2.5-StepProver, and Goedel Prover series formalize vast Lean corpora for expert iteration (Xin et al., 2024a; Wu et al., 2024; Lin et al., 2025a;b), while others generate synthetic formal proofs without informal seeds (Wu et al., 2020; Wang & Deng, 2020; Poesia et al., 2024). Projects such as MMA, Lean-STaR, TheoremLlama, and Lean Workbook create natural-language/formal-language (NL–FL) pairs to mitigate data scarcity (Jiang et al., 2023; Lin et al., 2024; Wang et al., 2024; Ying et al., 2024); yet LLM brittleness still introduces subtle logical errors. Recent LLM tools embedded in user workflows (e.g., LeanDojo, LeanCopilot, LLM-Step) further streamline manual formalization by suggesting premises and proof tactics (Yang et al., 2023; Song et al., 2023; Welleck & Saha, 2023; Shen et al., 2025). Our work tackles the complementary challenge of autoformalization: synthesizing intent-rich, user-style queries from formal Lean statements to better serve practitioners. Unlike previous informalization approaches that simply translate formal statements into natural language, our pipeline explicitly models user intent and query phrasing. Injecting this intent-aware signal during fine-tuning yields significant retrieval improvements on real user queries, and bridges the gap between informal user needs and the formal knowledge encoded in Lean libraries.

## B    DETAILS FOR DATASET CONSTRUCTION

### B.1    DATA SOURCE DETAILS

To build a more comprehensive and representative corpus of Lean 4 statements, we consider the following sources in addition to mathlib4:

- **Research-linked repository:** GitHub projects that are part of math papers, which often contain formalizations of novel theorems and lemmas.

- **Domain-specific library:** Projects such as SciLean that cover applied mathematical domains.

- **Transitive dependency:** Any Lean 4 project dependencies required by the above repositories.

We use LeanDojo (Yang et al., 2023) to extract formal statements from collected projects. The complete list of repositories in our data is shown in Table 5.

Table 5: Lean sources included in our dataset beyond mathlib4.

| Project name | Category | URL |
|---|---|---|
| Mathlib4 | Domain-specific library | https://github.com/leanprover-community/Mathlib4 |
| SciLean | Domain-specific library | https://github.com/lecopivo/SciLean |
| A formalization of Borel determinacy in Lean Manthe (2025) | Research-linked repository | https://github.com/sven-manthe/A-formalization-of-Borel-determinacy-in-Lean |
| Optlib Li et al. (2025a;b) | Research-linked repository | https://github.com/optsuite/optlib |
| Formalising Fermat's Last Theorem for Exponent 3 Monticone (2024) | Research-linked repository | https://github.com/riccardobrasca/flt3 |
| Formalisation of constructable numbers Monnerjahn (2024) | Research-linked repository | https://github.com/Louis-Le-Grand/Formalisation-of-constructable-numbers |
| Plausible | Transitive dependency | https://github.com/leanprover-community/plausible |
| ProofWidgets4 | Transitive dependency | https://github.com/leanprover-community/ProofWidgets4 |
| Aesop | Transitive dependency | https://github.com/leanprover-community/aesop |
| Batteries | Transitive dependency | https://github.com/leanprover-community/batteries |

## B.2 SUMMARY OF OUR DATASET

We summarize our dataset by input modalities in Table 6 and by data sources in Table 7.

Table 6: Overview of our dataset by input modalities.

| Input Modality | Samples | Percent |
|---|---|---|
| Synthesized User Query | 582102 | 42% |
| Augmented Proof State | 337647 | 24% |
| Informalized Statements | 244521 | 17% |
| Formal Statement | 244521 | 17% |

Table 7: Overview of our data sources. Note that the numbers below are *original* samples collected, and the numbers in Table 6 indicate *augmented* samples.

| Source | Samples | Percent |
|---|---|---|
| Mathlib | 238021 | 97% |
| Research-linked repository | 2198 | 1% |
| Lean 4 related libraries | 4302 | 2% |

## B.3 INFORMALIZATION DETAILS

The context we provide to GPT-4o consists of the following six components:

- **Formal name:** The full name of the statement to be informalized.
- **Formal statement code:** The complete Lean 4 code, including the statement and body.
- **Docstring:** Any human-written docstring associated with the statement in the codebase.
- **Neighbor statement:** The closest statement in the same file, based on positional proximity. We include its full code, as nearby statements often share related concepts. This helps the model understand the local context and inter-connectiness of the statement.
- **Dependent statements:** All the dependent statements used in the body of the statement to be informalized, along with their informalizations. This provides the necessary prerequisite knowledge and semantic dependencies for achieving high-quality informalization.

- **Related statement in Herald** (Gao et al., 2024b): A formal statement and its corresponding informalization from the Herald dataset that closely aligns with the target statement. This example serves as an in-context demonstration for GPT-4o, helping it generate a more accurate and stylistically consistent informalization. We use DeepSeek-Prover to retrieve the most relevant statements from Herald, and use the top-matching informal statement along with its corresponding formal statement as the related statement in the prompt for informalization.

### B.4 Augmented Proof State Details

We introduce an augmented proof-state input modality, where a proof state in Lean denotes the current list of pending subgoals, each with its local context of hypotheses and a target proposition. Some previous search engines support queries of type proof state (Tao et al., 2025a; Shen et al., 2025; Yang et al., 2023) to retrieve relevant theorems that could be applied in the next step, but they neglect that from a given proof state, we can expect multiple plausible directions to advance the proof. A mathematician might therefore want to search not only from the proof state itself, but also conditioned on their intended path forward.

Formally, a theorem's proof trajectory $\pi$ with $m$ steps can be represented as a sequence of states and tactics

$$\pi = \left\langle (S^{(0)}, T^{(1)}, S^{(1)}), \ (S^{(1)}, T^{(2)}, S^{(2)}), \ \ldots, \ (S^{(m-1)}, T^{(m)}, S^{(m)}) \right\rangle$$

where $S^{(0)}$ is the initial proof state, each $T^{(i)}$ is a tactic, and $S^{(m)}$ denotes no remaining goals (proof completed). Each triplet $(S^{(i-1)}, T^{(i)}, S^{(i)})$ corresponds to a transition step. From these trajectories, we collect unique state-transition pairs $(S^{(i-1)}, S^{(i)})$ where the tactic $T^{(i)}$ makes use of at least one premise (i.e., a previously established Lean statement).

To support this finer-grained retrieval, we synthetically generate augmented proof states: given a state-before and state-after pair $(S^{(i-1)}, S^{(i)})$ collected from proofs of Lean statements, we prompt GPT-4o to produce a natural description of the intended transition, without revealing the exact theorems used in $T^{(i)}$. This augmentation better supports users querying with proof states, enabling them to specify how they intend to proceed and to receive theorem suggestions that are more precisely aligned with their reasoning process. Lean Finder also implicitly supports raw proof-state inputs, ensuring compatibility with conventional retrieval settings while offering more fine-grained capabilities.

### B.5 Discussion on Imbalanced Clusters in Synthetic User Queries

We note a clear cluster imbalance within the synthetic user queries modality, as shown in Figure 4. For instance, the Metaprogramming cluster accounts for only 1% of the data, whereas Lemma Search and Proof Engineering dominate with 41% each. This imbalance arises from the filtering step applied prior to query generation (Section 3.1.2), where we ensure that each Lean statement is genuinely answerable under the chosen cluster. As a result, synthetic queries reflect natural usage patterns rather than being artificially uniform across categories. This skew is not an artifact of our pipeline alone and a similar distribution emerges in real user behavior. On Lean Zulip Chat, the *metaprogramming/tactics* channel sees roughly 13 messages per week, whereas the estimated count of messages per week in the channel *is there code for X* exceeds 300. Thus, the scarcity of metaprogramming queries relative to lemma search mirrors genuine community practices, reinforcing that our synthetic data distribution aligns with the way mathematicians actually interact with Lean.

## C OpenAI API Configurations

We prompt OpenAI models during data collection and generation, and we describe the detailed configurations below.

**Informalization.** We leverage the `GPT-4o` API to convert formal Lean statements into informal statements (Section 3.2). The temperature is set to 0.2 and the maximum output token limit is 1000.

Inspired by Herald Gao et al. (2024b), the prompt for informalization is shown in Section O.1.

**User Query Filtering.** We employ `GPT-4o` to filter out user queries that cannot be addressed by any Lean 4 formal statement (Section 3.1.1). The temperature is set to 0.0 with a maximum of 500 output tokens. The prompt for query filtering is shown in Section O.2.

**User Intent Clustering.** To cluster user intents (Section 3.1.1), we use the `o3` API. The process involves two stages:

- **Bootstrap Categorization.** An initial set of representative user queries is clustered using a temperature of 1.0 and a token limit of 20000. The prompt used for initial cluster generation is shown in Section O.3.
- **Progressive Clustering.** Remaining queries are progressively categorized with the same temperature and token settings. The prompt used for this part is shown in Section O.4.

**User Query Generation.** To synthesize diverse user queries (Section 3.1.2), we use the `GPT-4o` API with a temperature of 0.7 and a maximum output of 1000 tokens. The prompt used is shown in Section O.5.

## D    TRAINING DETAILS

We fine-tune our Lean Finder based on the DeepSeek-Prover-V1.5-RL 7B Xin et al. (2024b), using a query–code contrastive learning objective. We apply LoRA Hu et al. (2021) with a rank $r = 64$ and $\alpha = 128$. We train Lean Finder for 1 epoch with a per-GPU batch size of 8, gradient accumulation steps of 4, and 4 NVIDIA 6000Ada GPUs. We use the AdamW optimizer with a learning rate of $2 \times 10^{-5}$ and $\epsilon = 1 \times 10^{-8}$. The learning rate is linearly warmed up over the first 1550 steps, then decayed to 0 using a cosine schedule. The group size $G$ is 8 and the temperature $\tau$ is set to 0.01.

For DPO, we start with the checkpoint trained with contrastive loss described above. The two losses $\mathcal{L}_{\text{DPO}}$, $\mathcal{L}_{\text{contrastive}}$ are computed on *independent batches*, where the DPO loss $\mathcal{L}_{\text{DPO}}$ is trained on preference data from user and LLM feedback and the contrastive loss $\mathcal{L}_{\text{contrastive}}$ is trained on a mixture of the original training set (which is more abundant and less noisy) and new user queries. This design mitigates the limitations of real feedback data, which are relatively scarce, noisy, and potentially shifted in distribution compared to the original input modalities. We train with 4 NVIDIA 6000Ada GPUs using a per-GPU batch size of 2 and 4 gradient accumulation steps for a total of 3 epochs. We set $\beta$ to 0.1 and $\lambda$ to 0.01.

## E    PREFERENCE DATA CONSTRUCTION DETAILS

Our preference dataset consists of triplets $(q, c^+, c^-)$, where $q$ is a user query and $c^+$ and $c^-$ are, respectively, a chosen and rejected Lean statement. We construct these triplets from three complementary sources:

- **Direct retrieval-level feedback.** Users may upvote or downvote individual retrieved statements. For a query $q$, we form triplets by pairing each upvoted statement with each downvoted one. This provides explicit statement-level supervision from user feedback. See Figure 9 for an example.
- **Model-level feedback.** We also design blinded comparison interface on our web service where users are shown the top-$k$ results for a given query $q$ from two retrievers Lean Finder ($\mathcal{R}_{\text{LF}}(q)$) and Lean Search ($\mathcal{R}_{\text{LS}}(q)$) without knowing which system produced which results as the order of the retrievers is randomized across interactions, ensuring unbiased comparisons (See Figure 9 for an example). Users then select one of four outcomes: *Retriever A better*, *Retriever B better*, *Tie*, or *Both bad*. Since this feedback is coarse, we refine it into triplets using GPT-4o with rules to best preserve user feedback. For example, if Lean Finder is preferred, then at least one statement in the set difference $\mathcal{R}_{\text{LF}}(q) \setminus \mathcal{R}_{\text{LS}}(q)$ is labeled helpful and at least one in $\mathcal{R}_{\text{LS}}(q)$ is labeled unhelpful; ties require at least one helpful statement in the intersection $\mathcal{R}_{\text{LF}}(q) \cap \mathcal{R}_{\text{LS}}(q)$; both-bad cases are discarded. For each query, every selected $c^+$ is paired with $c^-$ to form $(q, c^+, c^-)$ triplets. As we use GPT-4o in the process of constructing preference data, we evaluate how well GPT-4o aligns

with human preference (Appendix H). The results show that GPT-4o aligns well with human preference.

- **Lean Zulip discussions.** To diversify queries beyond our web service, we extract real user questions from Lean Zulip. Since these lack explicit statement-level feedback, GPT-4o is prompted to identify helpful and unhelpful retrieved statements from Lean Finder. This source captures community-driven problem formulations.

The resulting dataset $\mathcal{P} = \{(q_i, c_i^+, c_i^-)\}_{i=1}^{N}$ aggregates these sources. By combining explicit statement-level votes, blinded system-level comparisons with refinement, and original community queries, we construct a heterogeneous preference resource for aligning with user preference in Lean retrieval.

## F    USER STUDY DETAILS

To evaluate the effectiveness of Lean Finder in handling real-world user queries, We conduct a user study with 5 participants using 128 high-quality real user queries for Lean statements, collected from GitHub. The participants all have rich experiences in Lean 4, with two PhD holders in Mathematics and Computer Science, a Master's student in Physics, and two Undergraduate students majoring in Mathematics.

For each query in the user study, we retrieve the top-3 results from each of the models. The identity of the models is hidden from participants, and the order of models is randomized for every query to ensure fairness. Because GPT-4o can hallucinate incorrect Lean statements that do not exist in mathlib or other Lean libraries, we apply a special treatment: instead of prompting it to generate full Lean statements directly, GPT-4o is asked to produce only the full name of the statement it believes answers the query. We then match this name against our database of mathlib statements and retrieve the corresponding ground-truth statement. This guarantees that GPT-4o's results are valid Lean statements and prevents hallucinated outputs from affecting the evaluation.

Participants are asked to evaluate the retrievals by selecting and ranking the top three models that provide the most helpful results for answering the given query. To account for ties, participants may assign the same rank (Best, Second Best, or Third Best) to multiple models. If none of the retrieved results are useful, or if the participants don't know how to answer the query, they may indicate this through a dedicated option.

Examples of the user study format is in Figure 6.

## G    DETAILS FOR EVALUATION SETTINGS AND DATA

We describe below the details of evaluation settings and data we used for each input modality.

**Evaluation on Informalized Statements.**    Since GPT-4o is included as a baseline but is not a retrieval model and lacks direct access to mathlib and other Lean libraries during inference, we enable its web-search tool and restrict its output to ensure a fair comparison. Specifically, GPT-4o is prompted to generate only the *full name* of a Lean statement (the fully qualified identifier with namespace hierarchy), which avoids the generation of hallucinated Lean code. We evaluate GPT-4o under two matching strategies:

1. **Full-name match:** the prediction is correct only if the full name exactly matches the ground truth.
2. **Stem match:** a looser criterion where the prediction is correct if the name stem (e.g., `coeAlgEquiv` in `DirectSum.IsInternal.coeAlgEquiv`) matches.

For all other models, a retrieved statement is considered correct only if the full Lean code matches.

To ensure fair comparison with Lean Search, which has a fixed database, we impose two constraints: 1) The ground-truth formal statement *must* exist in the Lean Search database (otherwise Lean Search would always fail to retrieve it). 2) The corresponding informal query *must not* appear in either the Lean Search database or our training data (otherwise Lean Search would retrieve it perfectly, and our model would already have seen a near-duplicate). Following these rules, we construct a fair test set

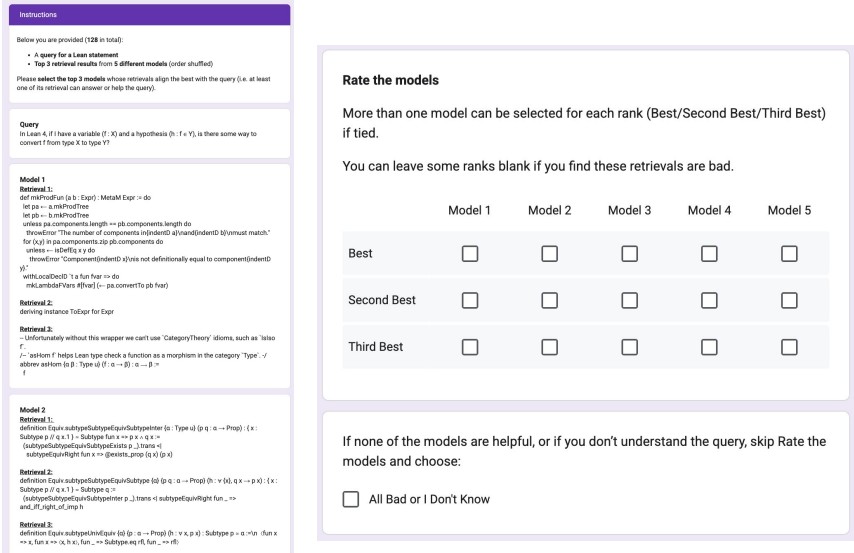

Figure 6: Google Form for the user study. We compare five models: Lean Finder, Lean Search (Gao et al., 2025), GPT-4o, Lean Finder w/o DPO, and Lean Finder w/o DPO and synthetic user query. Each model retrieves 3 Lean statements, with the model order randomized across questions to avoid bias.

by randomly sampling 1000 Lean statements from Lean Search's database that are not in our training set and informalize them with a simplified prompt.

**Formal Statement.**    For the statement input modality, we use 1000 Lean statements from the Lean Search database, ensuring none appear in Lean Finder's training set. To simulate noisy queries and test model robustness, we randomly replace 20% of the tokens in each statement with tokens sampled from the vocabulary. To prevent invalid characters that may arise from the retrieval model's tokenizer during augmentation, we instead use a custom tokenizer. The same matching strategies for GPT-4o is applied.

**Synthetic User Query.**    For the synthetic user query modality, we use 1000 queries from our test set whose ground truth statement is in Lean Search database. Same matching strategies for GPT-4o are used to compare with Lean Finder in this input modality.

**Proof State.**    Finally, for the proof state modality, we randomly sample 2224 proof states from the test set. We evaluate both the augmented proof state queries (the modality that Lean Finder is explicitly trained on) and the raw proof state queries. Same matching strategies for GPT-4o are used to compare with Lean Finder in this input modality.

## H    EVALUATING GPT-4O ALIGNMENT WITH HUMAN PREFERENCES

Since GPT-4o is used to extract preference pairs for constructing our DPO fine-tuning data, we evaluate how well it aligns with human preferences on Lean statements through two complementary tests.

**Retriever-level preference.**    We use 100 queries collected from our web service, where users compared retrieval results from Lean Finder and Lean Search and could choose one retriever as the better one, a tie, or both bad. Since the ground-truth preferences are known from users, we directly prompt GPT-4o to make a choice from the same four options. For the 63 cases where users preferred one retriever over the other, GPT-4o match the human choice 58 times (92%). However, in the 37 cases where users selected *tie* or *both bad*, GPT-4o is correct only 15 times (43%). Manual inspection

of GPT-4o's explanations suggests that the model tends to force a choice between retrievers, often providing far-fetched reasoning instead of recognizing the *tie* or *both bad* scenarios.

**Statement-level preference.** We also evaluate 35 unique queries paired with user feedback on individual statements (upvotes or downvotes). Since users may not comprehensively rate every statement they see, we use a *coverage* metric: GPT-4o is asked to classify statements as useful or not useful for answering the query, and we check whether all statements that received human feedback receive the same label as predicted by GPT-4o. We find that in 31 out of 35 queries, GPT-4o achieve full coverage, i.e., every user-rated statement matches GPT-4o's judgment.

These results indicate that GPT-4o closely mimics human preferences at both the retriever and statement levels, which indicates it's strong understanding in Lean semantics. While the model struggles to recognize *tie* or *both bad* cases, these scenarios are not relevant to our preference data collection process. Overall, this confirms that GPT-4o is a reliable tool for extracting preference pairs aligned with human judgment.

Despite its strength in preference alignment, GPT-4o performs poorly on the retrieval task from Section 4.1. In the retrieval setting, the model is required to take a natural user query and, with access to a web search tool, output the full name of the relevant Lean statement. While GPT-4o demonstrates a strong understanding of Lean semantics and can interpret the formal statements, it struggles to locate the exact corresponding result within the vast space of mathlib and the broader internet. The challenge is in precise identification: natural queries often map to highly specific statements which can be hard to locate through open-ended search. This makes retrieval particularly demanding for GPT-4o, in contrast to specialized systems like Lean Finder that are designed to index and surface the correct statement reliably.

# I  RAG FOR LLM PROVERS DETAILS

To examine whether Lean Finder can act as a prover-agnostic retrieval backbone, we integrate it with two representative types of LLM-based theorem provers: whole-proof generation model (Section I.1) and step-wise theorem proving model (Section I.2). The whole-proof generation models take a theorem statement and produces an entire proof in one shot, while step-wise provers interact with the proof assistant iteratively, generating one step at a time based on the current proof state. Each integration follows a retrieval-augmented generation (RAG) setup, where Lean Finder provides relevant supporting lemmas that the prover can condition on during proof generation.

## I.1  WHOLE PROOF GENERATION MODEL

For whole-proof generation provers, we evaluate Goedel Prover (Lin et al., 2025a) and DeepSeek-Prover V1.5 (Xin et al., 2024b). Experiments are conducted on three benchmarks: MiniF2F (Zheng et al., 2022), ProofNet (Azerbayev et al., 2023), and the PutnamBench (Tsoukalas et al., 2024). For each theorem to be proved, we first pass the initial proof state of the theorem to Lean Finder, which retrieves between six and ten relevant statements from our Lean statement corpus that are likely to be helpful in completing the proof. These retrieved statements are added to the prover's input as supporting context.

The proof performance with and without Lean Finder is summarized in Table 8. Integrating Lean Finder retrieval into whole-proof generation provers yields modest but consistent improvements on MiniF2F and ProofNet. Although the gains are relatively small, the positive shift indicates that Lean Finder is compatible with existing provers and can provide helpful supporting context without disrupting performance. This result is encouraging given our primary goal of building a user-centered search engine that effectively captures user intent. On the more challenging PutnamBench dataset, however, there is inconsistent performance difference between using Lean Finder and not using it, suggesting that current provers are not yet able to fully leverage retrieved statements for highly complex and creative problems. Overall, these results demonstrate that Lean Finder integrates seamlessly with provers while preserving stability and delivering measurable benefits on standard benchmarks.

Table 8: Performance of whole-proof generation models on MiniF2F, ProofNet, and PutnamBench, comparing with and without Lean Finder retrieval (LF: integration with Lean Finder retrieval). The best results are presented in **bold**.

| Prover System | Budget | MiniF2F Test | | ProofNet Test | | PutnamBench | |
|---|---|---|---|---|---|---|---|
| | | w/o LF | w/ LF | w/o LF | w/ LF | w/o LF | w/ LF |
| Goedel-Prover-SFT | 1 | 38.1% | **38.5%** | 6.6% | **7.2%** | **3** | 2 |
| | 32 | 57.7% | **58.2%** | 13.2% | **14.4%** | 6 | 6 |
| | 128 | 59.4% | **59.8%** | 17.6% | 17.6% | 7 | 7 |
| DeepSeek-Prover-V1.5-RL | 1 | 34.4% | **35.7%** | 4.3% | **6.5%** | 4 | **5** |
| | 32 | 50.0% | 50.0% | 14.5% | **18.3%** | **7** | 6 |
| | 128 | 51.6% | 51.6% | 17.7% | **19.4%** | 7 | 7 |

## I.2 STEP-WISE THEOREM PROVING MODEL

For step-wise provers, we focus on REALProver (Shen et al., 2025), which consists of a retrieval module and a tactic generator. The tactic generator is trained to produce the next proof step conditioned on the retrieved statements provided by the retrieval module. In our setup, we replace REALProver's original retrieval module with Lean Finder. At each proof step, Lean Finder receives the current proof state and retrieves candidate statements, which are then fed into the tactic generator to generate the next tactic.

We evaluate this setup on three benchmarks: MiniF2F (Zheng et al., 2022), ProofNet (Azerbayev et al., 2023), and FATE-M (Shen et al., 2025) which is the dataset introduced in the REALProver. The proof performance with Lean Finder integration is summarized in Table 9.

Table 9: Performance of REALProver on MiniF2F, ProofNet, and FATE-M test sets, comparing with and without Lean Finder retrieval (LF = integration with Lean Finder retrieval). The best results are in **bold**.

| Prover System | Budget | MiniF2F | | ProofNet | | FATE-M | |
|---|---|---|---|---|---|---|---|
| | | w/o LF | w/ LF | w/o LF | w/ LF | w/o LF | w/ LF |
| REALProver | 8×8 | 52.0% | 52.0% | **23.7%** | 23.1% | **51.1%** | 49.6% |

The results in Table 9 show that integrating Lean Finder into REALProver maintains largely similar performance, with negligible changes across datasets. This suggests that REALProver's performance may be more constrained by its tactic generator than by retrieval. Moreover, its tactic generator may be tuned to the specific retrieval signals of its native module, so replacing it with Lean Finder does not yield immediate gains.

The limited gains suggest that while Lean Finder can be integrated into additional provers, the immediate benefits are marginal, which is why we restrict our experiments to a few representative systems.

## J ADDITIONAL EXPERIMENTS

We further compare Lean Finder with Lean Search that has augmentation mode enabled, Qwen3-Embedding-8B (Zhang et al., 2025), and Lean Explore (Asher, 2025) on their supported input types. For Lean Search with augmentation mode enabled, we first use its built-in query augmentation functionality to augment the query, then submit it to Lean Search for retrieval. Since Lean Search and Lean Explore doesn't support proof state as input, we do not compare their retrieval performance when the input is of type proof state. We require the ground truth formal statement to be present in the respective static databases for both Lean Search and Lean Explore for fair comparison. Because this filtering process yields different test sets for each, we report their results separately.

Lean Finder consistently surpasses all baselines, including specialized Lean search engines (such as Lean Explore) and the state-of-the-art Qwen3-Embedding-8B model, which is larger than Lean Finder by 1B parameters. This indicates Lean Finder's strong performance and robustness across input types.

Table 10: Retrieval performance comparison between Lean Finder, Lean Search with augmentation mode enabled, and Qwen3-Embedding-8B across three input modalities: informalized statement, synthetic user query, and augmented statement.The best results are presented in **bold**.

| Model | Informalized Statement | | | | Synthetic User Query (Sec. 3.1) | | | | Augmented Statement | | | |
|---|---|---|---|---|---|---|---|---|---|---|---|---|
| | R@1 | R@5 | R@10 | MRR | R@1 | R@5 | R@10 | MRR | R@1 | R@5 | R@10 | MRR |
| Lean Finder | **64.2** | **88.9** | **93.3** | **0.75** | **54.4** | **84.4** | **91.4** | **0.68** | **82.7** | **97.0** | **97.7** | **0.89** |
| Lean Search (augmentation mode) | 31.9 | 77.4 | 84.8 | 0.44 | 30.0 | 72.7 | 83.3 | 0.43 | 45.4 | 86.0 | 93.0 | 0.58 |
| Qwen3-Embedding-8B | 57.7 | 82.5 | 89.2 | 0.69 | 41.2 | 73.8 | 82.8 | 0.55 | 76.2 | 93.4 | 95.1 | 0.83 |

Table 11: Comparison of Lean Finder with Qwen3-Embedding-8B on retrieval performance across two input modalities: augmented proof state and raw proof state. "Augmented proof state": proof state augmented with natural language description of proofs (Section 3.2). "Raw proof state": proof state without the augmentation. The best results are presented in **bold**.

| Model | Augmented Proof State | | | | Raw Proof State | | | |
|---|---|---|---|---|---|---|---|---|
| | R@1 | R@5 | R@10 | MRR | R@1 | R@5 | R@10 | MRR |
| Lean Finder | **24.6** | **56.8** | **67.9** | **0.40** | **8.3** | **30.1** | **40.0** | **0.19** |
| Qwen3-Embedding-8B | 8.0 | 26.1 | 34.7 | 0.17 | 5.4 | 21.1 | 27.4 | 0.13 |

Table 12: Retrieval performance comparison between Lean Finder and Lean Explore on the input types that Lean Explore support: informalized statement, synthetic user query, and augmented statement. The test set used is a cleaned version of the original test set to ensure the ground truth Lean statement is present in Lean Explore's static database.The best results are presented in **bold**.

| Model | Informalized Statement | | | | Synthetic User Query (Sec. 3.1) | | | | Augmented Statement | | | |
|---|---|---|---|---|---|---|---|---|---|---|---|---|
| | R@1 | R@5 | R@10 | MRR | R@1 | R@5 | R@10 | MRR | R@1 | R@5 | R@10 | MRR |
| Lean Finder | **65.7** | **89.0** | **93.2** | **0.76** | **57.3** | **85.4** | **91.2** | **0.69** | **86.8** | **96.7** | **98.0** | **0.91** |
| Lean Explore | 35.0 | 61.8 | 68.0 | 0.46 | 26.3 | 51.3 | 60.1 | 0.37 | 85.8 | 93.6 | 95.2 | 0.89 |

## K  ABLATION STUDIES

**DPO improves alignment with human preference.**   Table 13 shows that adding DPO improves retrieval quality on real user queries. Compared to the variant trained without DPO, Lean Finder is more frequently chosen by participants as the preferred model in the user study. This supports our design goal: DPO fine-tuning on preference pairs mined from user and LLM feedback yields a retriever that better aligns with human judgments, even though the underlying retrieval metrics on synthetic data degrade only slightly as shown in Table 14.

**Synthetic queries are essential for intent understanding.**   To isolate the effect of synthetic user queries, we train a variant without both DPO and synthetic queries while keeping training steps consistent. In the user study (Table 13), this model receives lower Best and Second Best votes compared to both the full Lean Finder and the variant trained without DPO. Since the only difference compared to the latter is the absence of synthetic queries during training, this decline provides strong evidence that synthetic queries are critical for teaching the model how to interpret natural, free-form user inputs. The user study thus offers the clearest demonstration that synthetic queries are critical for helping the retriever to understand real user intent. In retrieval metrics on synthetic data (Table 14 and Table 15), we also observe a moderate decrease in performance for input modalities closer to natural queries like informalized statement and synthetic user queries, while more structured modalities like proof states remain largely unaffected, reflecting the fact that synthetic queries rarely involve proof states.

**Alignment–retrieval trade-off.**   In Table 14, we compare Lean Finder with the variant trained without DPO on retrieval when the inputs are synthetic data. We observe a small but consistent reduction in retrieval metrics across most modalities after DPO fine-tuning. We interpret this as a modest *alignment tax*: even with a regularized objective that combines contrastive and DPO losses, shifting the embedding space to match human preferences necessarily incurs a slight cost in retrieval metrics. However, this trade-off is offset by the clear gains in the user study (Table 13), where the DPO-aligned model demonstrates improved usability and better alignment with real user preferences.

Due to the time and financial costs of semantic analysis of real-world queries (Section 3.1.1), we do not explore different number of intent clusters by re-clustering real user queries.

Table 13: Comparison of Lean Finder and it's variants on the user study. *w/o DPO* is the model fine-tuned without DPO. *w/o SynQ+DPO* is the model fine-tuned without synthetic user query and without DPO. $n_1(m)$, $n_2(m)$, $n_3(m)$ refers to the counts of 1st/2nd/3rd place votes for model $m$. "SynQ": our synthetic user query.

| Model ($m$) | $n_1(m)$ | $n_2(m)$ | $n_3(m)$ | Top–3 (%) | Borda$_{norm}$ |
|---|---|---|---|---|---|
| Lean Finder | **139** | **56** | 36 | **81.6%** | **0.67** |
| w/o DPO | 132 | 51 | 34 | 76.7% | 0.63 |
| w/o SynQ+DPO | 90 | 51 | **37** | 62.9% | 0.48 |

Table 14: Comparison of Lean Finder and it's variants on retrieval performance where the input modalities are informalized statement, synthetic user query, and augmented statement. *w/o DPO* is the model fine-tuned without DPO. *w/o SynQ+DPO* is the model fine-tuned without synthetic user query and without DPO. The best results are presented in **bold**. "SynQ": our synthetic user query.

| Model | Informalized Statement | | | | Synthetic User Query | | | | Augmented Statement | | | |
|---|---|---|---|---|---|---|---|---|---|---|---|---|
| | R@1 | R@5 | R@10 | MRR | R@1 | R@5 | R@10 | MRR | R@1 | R@5 | R@10 | MRR |
| Lean Finder | 64.2 | 88.9 | 93.3 | 0.75 | 54.4 | 84.4 | 91.4 | 0.68 | 82.7 | 97.0 | 97.7 | 0.89 |
| w/o DPO | **66.9** | **91.2** | **95.1** | **0.77** | **58.1** | **86.8** | **93.1** | **0.71** | 84.9 | **97.4** | **98.4** | **0.91** |
| w/o SynQ+DPO | 61.1 | 87.5 | 91.6 | 0.73 | 38.4 | 66.4 | 73.7 | 0.51 | **86.4** | 96.8 | 98.0 | **0.91** |

Table 15: Comparison of Lean Finder and it's variants on retrieval performance where the input modalities are augmented proof state and raw proof state. *w/o DPO* is the model fine-tuned without DPO. *w/o SynQ+DPO* is the model fine-tuned without DPO and without synthetic user query. The best results are presented in **bold**. "SynQ": our synthetic user query.

| Model | Augmented Proof State | | | | Raw Proof State | | | |
|---|---|---|---|---|---|---|---|---|
| | R@1 | R@5 | R@10 | MRR | R@1 | R@5 | R@10 | MRR |
| Lean Finder | 24.6 | 56.8 | 67.9 | 0.40 | 8.3 | **30.1** | 40.0 | **0.19** |
| w/o DPO | 27.7 | 62.3 | **74.1** | **0.44** | **8.8** | 29.5 | 39.9 | **0.19** |
| w/o SynQ+DPO | **28.5** | **62.5** | 73.7 | **0.44** | 8.7 | 28.4 | **41.0** | 0.18 |

## L  MODULE-WISE ANALYSIS

Figure 7 shows Lean Finder's recall@1 across mathematical modules, comparing synthetic user queries (blue) with informalized statements (orange). Performance is consistently higher with informalized statements, reflecting the stronger alignment between formal statements and their informalized counterparts than with free-form queries. Modules such as Combinatorics, Ring Theory, and Probability achieve particularly high recall with informalization, exceeding 80–90%. By contrast, synthetic user queries yield lower recall overall, especially in Set Theory, Ring Theory, and Data, where recall dips near 40%. The gap highlights the challenge of bridging natural queries with formal statements.

## M  LEAN FINDER'S USER INTERFACE

To facilitate Lean users, we also develop and deploy our user interface, as shown in Figure 8 and Figure 9. We retrieve formal statements and also show their informalizations to help users better understand the Lean statements. Our Lean Finder also includes an Arena mode (Figure 9), where users can compare retrieval results from Lean Finder and Lean Search. The two retrieval systems are presented in randomized order as Retriever A or Retriever B, and users may vote for the better one, choose a tie, or mark both as bad. These pairwise preferences, together with votes on individual retrieved statements, form the core of our DPO fine-tuning data.

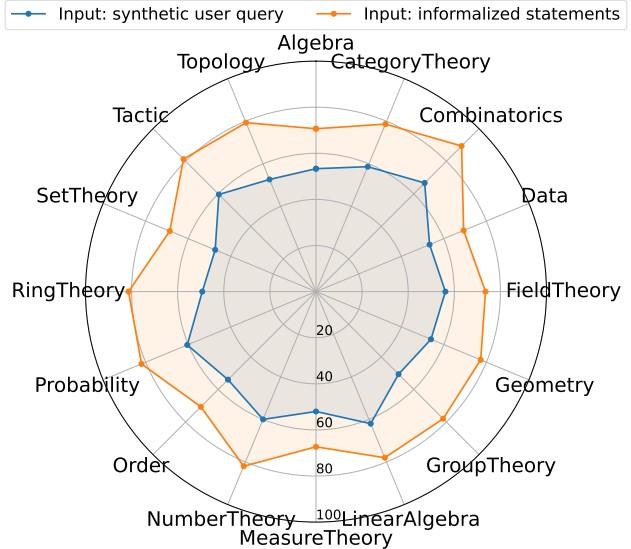

Figure 7: Lean Finder recall@1 performance on different mathematical modules for synthetic user query and informalized statements as inputs.

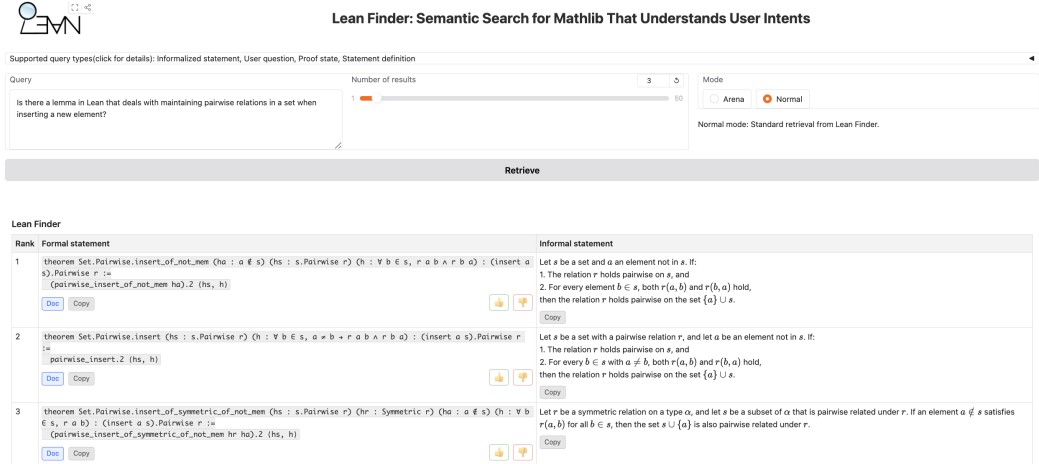

Figure 8: Web service interface of our Lean Finder in the **Normal** mode.

## N  PRIVACY AND DATA RELEASE

To protect user privacy, we do not release any real user discussion data collected from Lean Zulip. We will only release these synthetic queries, along with queries from other input modalities. No identifiable information or original messages from Zulip users are included in our datasets.

## O  PROMPTS

### O.1  PROMPT USED FOR LEAN STATEMENT INFORMALIZATION

```
You are an expert mathematician and an expert in Lean and Mathlib.

**Instruction**: Your task is to translate the formal theorem below
```

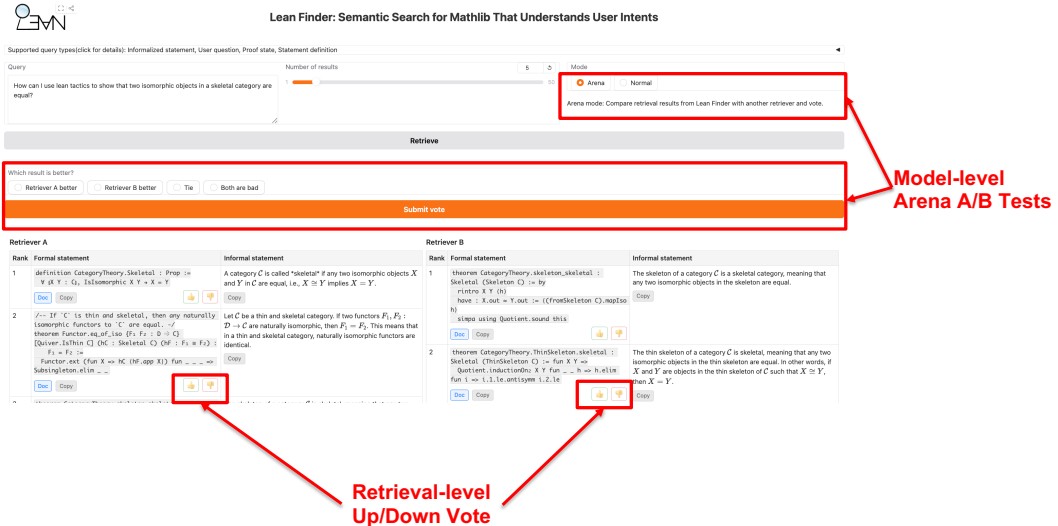

Figure 9: Web service interface of our Lean Finder in the **Arena** mode. We collect both the retrieval-level and model-level feedback, which supports the fine-tuning of Lean Finder.

```
into an informal statement (in LaTeX) that is more accessible to
mathematicians.
Inputs include:
1. Formal name: ...
2. Formal statement: ...
3. Docstrings: ...
4. Neighbor statement: ...
5. Dependent theorems: ...
6. Related statement: ...
Then create an informal name and informal statement.

**Principles of Informal Statements**
Use human mathematical notations and LaTeX formulas as much as
possible
...

**Principles of Informal Names**:
Emphasize logical relationships, not just listing concepts.
...

Input:
*Formal name*:
*Formal statement*:
*Docstring*:
*Neighbor statement*:
*Dependent theorems*:
*Related theorem*:
```

## O.2 PROMPT USED FOR USER QUERY FILTERING

```
You are an expert at Lean 4 and an experienced user of Lean Zulip Chat
.
```

```
**Instruction**:Your task is to first read the excerpt provided below
that contains the first 5 messages from a discussion thread in Lean
Zulip Chat and then decide to accept or reject the excerpt based on
the following criteria:
1. You should accept the excerpt if the main user question in the
excerpt can be answered at least partially by providing a Lean 4
statement.
2. You should accept the excerpt if showing a pre-existing Lean 4
statement will move the discussion forward in a meaningful way.
3. You should reject the excerpt if none of the above is true for this
 excerpt.
4. When uncertain, use your best judgment.

Then identify the main question in this discussion thread...

Examples (not exhaustive):
ACCEPT: Is there a lemma/theorem/definition that ..."?
...

REJECT: Library infrastructure, version bumps, build tools, CI, style.
...

**Input**:
```

## O.3  PROMPT FOR GENERATING INITIAL CLUSTERS OF USER QUERIES

```
You are an expert at Lean 4 and an experienced user of Lean Zulip Chat
.

**Instruction**:
You will be provided below with 50 excerpts that contain the first few
 messages from discussion threads in Lean Zulip Chat and the main user
 question that is being asked in that discussion thread. Each excerpt
has been filtered so that the main user question can be answered at
least partially by a statement written in Lean 4 or showing a pre-
existing Lean 4 statement will move the discussion forward in a
meaningful way.

You need to group the following 50 excerpts into clusters by their
query purpose/type, and follow the instructions below:
1. Each cluster should represent a distinct type of query based on the
 underlying intent or perspective of the main question in the excerpt.
2. Give each cluster a descriptive name (a few words) that summarizes
that query type.
3. For each cluster, give a detailed description of what the cluster
represents. This should include the common properties shared by the
queries that belong in these clusters, and the description of the
underlying intent or perspective of the queries made by the user. This
 must be detailed enough, such that a person reading this description
can create new queries for a Lean 4 statement that belong in this
cluster.
4. For each cluster, include at most 5 representative example queries
that best define this cluster in the description. The chosen example
queries must be the main_question provided in the input for the
excerpt.

The input will be a list of excerpts, where each excerpt will have the
 following contents:
1. channel: ...
2. topic: ...
3. main question: ...
```

```
4. messages: ...

**Input**:
```

## O.4    PROMPT FOR PROGRESSIVE CLUSTERING OF USER QUERIES

```
You are an expert at Lean 4 and an experienced user of Lean Zulip Chat
.

**Instruction**:
You will be provided below with 25 excerpts that contain the first few
 messages from discussion threads in Lean Zulip Chat and the main user
 question that is being asked in that discussion thread. Each excerpt
has been filtered so that the main user question can be answered at
least partially by a statement written in Lean 4 or showing a pre-
existing Lean 4 statement will move the discussion forward in a
meaningful way. You will also be provided with the current clusters of
 queries based on previous excerpts. Each cluster represents a
distinct type of query based on the underlying intent or perspective
of the main question in the excerpt. You need to do the following:

**Task A: Fitting new excerpts**:
1. For each input excerpt, try to fit the excerpt into an existing
cluster based on the 'clusters description and representative examples
 provided for queries in that cluster.
2. If new excerpts fit into an existing cluster of queries and the new
 excerpts reveal nuances that the current description of the cluster
does not capture, you should append at most one new sentence to an
existing cluster description.
3. If new excerpts fit into an existing cluster of queries and the
main_question in the new excerpts reveal nuances that the current
examples for that cluster does not capture, you should add at most one
 new example to an existing cluster examples.
4. DO NOT remove any description and examples in the existing cluster.
 You are only allowed to add new ones when needed.
5. If the excerpt DOES NOT fit into an existing cluster, do Task B,
otherwise skip Task B.

**Task B: New cluster proposal (if needed)**:
1. If the input excerpt 'doesnt fit into any existing cluster, then
you should propose one new cluster.
2. The new cluster must have a descriptive name and a detailed
description of what the cluster represents. The description should
include the property of the query in this new cluster, and the
description of underlying intent or perspective of the query made by
the user...
3. The new cluster must include at most 10 representative example
queries that best define this cluster. Each included example must
illustrate different situations within the cluster...here.
4. DO NOT remove existing clusters, even if they are not used by the
new excerpts.

Each input excerpt will have the following contents:
1. channel: ...
2. topic: ...
3. main question: ...
4. messages: ...

Each input cluster will have the following contents:
1. cluster_name: ...
2. cluster_description: ...
```

```
3. examples: ...

**Inputs**:
```

## O.5 PROMPT FOR GENERATING USER QUERIES

```
You are an expert Lean 4 user and an experienced participant in the
Lean Zulip chat. You know how people usually phrase questions when
they are searching for existing Lean statements.

**Instructions**:
You will be given:
1. The name, description, and 10 canonical examples of a specific
cluster of queries from Lean Zulip or GitHub. The queries in this
cluster share the same underlying intent or perspective, precisely
captured by the description.
2. A single Lean 4 formal statement, formal name, informal name, and
the natural language description of the statement.
Write one realistic user query that:
1. Clearly belongs in this cluster. The intent and perspective of the
query should mirror the description and examples.
2. Could be answered at least partially by showing the Lean 4
statement you were given, or would be moved forward in a meaningful
way by that statement.
3. Does not explicitly reveal the statement.
4. Feels like a genuine message a mathematician or Lean 4 user would
post (natural tone, sensible level of detail).

**Input**:
```

## P    DATASHEETS FOR DATASETS

This document is based on *Datasheets for Datasets* by (Gebru et al., 2021).

### P.1    MOTIVATION

**For what purpose was the dataset created?**  Was there a specific task in mind? Was there a specific gap that needed to be filled? Please provide a description.

The dataset was created to support user-centered semantic search in Lean by capturing the diverse intents of mathematicians. It includes synthesized user queries grounded in formal statements, informalized statements from various libraries and research-linked repositories, and augmented proof states. The purpose is to create retrieval models that better reflect real user needs and integrate seamlessly with LLM-based theorem provers. The dataset bridges real user queries and raw informalizations for search engines, fostering research in retrieval-augmented theorem proving and intent-aware formal reasoning.

**Who created this dataset (e.g., which team, research group) and on behalf of which entity (e.g., company, institution, organization)?**

The dataset was created by the anonymous authors of this Lean Finder project, affiliated with a research group focused on AI-for-math applications.

**What support was needed to make this dataset?**  (e.g.who funded the creation of the dataset? If there is an associated grant, provide the name of the grantor and the grant name and number, or if it was supported by a company or government agency, give those details.)

The creation of the dataset was supported by research funding for developing novel applications of LLMs in formal mathematics. Additional support included access to computational resources for fine-tuning and coverage of API costs for data synthesis and processing.

**Any other comments?**

N/A

### P.2    COMPOSITION

**What do the instances that comprise the dataset represent (e.g., documents, photos, people, countries)?**   Are there multiple types of instances (e.g., movies, users, and ratings; people and interactions between them; nodes and edges)? Please provide a description.

Each instance represents a Lean-related query, which includes statement pair or proof context designed to capture mathematicians' intents. The dataset includes multiple types of instances: (i) synthesized user queries grounded in formal Lean statements, (ii) informalized statements providing natural language descriptions of Lean theorems, (iii) augmented proof states representing transitions between proof steps, and (iv) formal statement extracted from Lean libraries and research-linked repositories. Together, these instance types reflect different ways mathematicians interact with Lean and enable comprehensive training for semantic retrieval models.

**How many instances are there in total (of each type, if appropriate)?**

The dataset comprises over 1.4 million query–code pairs. Specifically, it includes 582k synthesized user queries, 244k informalized statements, 338k augmented proof states, and 244k formal statement.

**Does the dataset contain all possible instances or is it a sample (not necessarily random) of instances from a larger set?**   If the dataset is a sample, then what is the larger set? Is the sample representative of the larger set (e.g., geographic coverage)? If so, please describe how this representativeness was validated/verified. If it is not representative of the larger set, please describe why not (e.g., to cover a more diverse range of instances, because instances were withheld or unavailable).

The dataset is a curated sample rather than the complete set of possible Lean queries. The larger set consists of all formal statements and proofs in mathlib4 and research-linked Lean repositories, along with the open-ended space of user queries posed by mathematicians. Since real user queries are

limited in number and often lack ground-truth resolutions, we synthesize queries grounded in formal statements to approximate this space. While not exhaustive, the dataset is designed to be broadly representative by covering multiple input modalities (synthetic user queries, informalizations, proof states, and formal statements) and by drawing from both mathlib and research-linked repositories to capture diverse mathematical content.

**What data does each instance consist of?** "Raw" data (e.g., unprocessed text or images) or features? In either case, please provide a description.
Each instance consists of raw text in the form of Lean 4 formal statements, their associated natural language variants, or proof contexts. Depending on modality, this may include: (i) synthesized user queries phrased in natural language, (ii) informalized statements produced from formal Lean code, (iii) augmented proof states describing proof progressions, and (iv) formal statement. All data are stored as text pairs.

**Is there a label or target associated with each instance?** If so, please provide a description.
Yes. Each query is paired with one or more target Lean 4 formal statements that serve as the ground-truth retrieval target. In the case of augmented proof states, the target can be one or more Lean statements used in the corresponding proof transition. For informalizations, synthetic user queries, formal statement, the target is the complete Lean statement from which they were derived.

**Is any information missing from individual instances?** If so, please provide a description, explaining why this information is missing (e.g., because it was unavailable). This does not include intentionally removed information, but might include, e.g., redacted text.
N/A

**Are relationships between individual instances made explicit (e.g., users' movie ratings, social network links)?** If so, please describe how these relationships are made explicit.
No. Each instance is an independent query–code pair.

**Are there recommended data splits (e.g., training, development/validation, testing)?** If so, please provide a description of these splits, explaining the rationale behind them.
Yes, the dataset is split into training and testing sets.

**Are there any errors, sources of noise, or redundancies in the dataset?** If so, please provide a description.
Yes. The synthesized queries from formal statements are generated using LLMs, some may contain minor phrasing inconsistencies or unnatural wording compared to real user queries. Informalizations may also result in inaccuracies.

**Is the dataset self-contained, or does it link to or otherwise rely on external resources (e.g., websites, tweets, other datasets)?** If it links to or relies on external resources, a) are there guarantees that they will exist, and remain constant, over time; b) are there official archival versions of the complete dataset (i.e., including the external resources as they existed at the time the dataset was created); c) are there any restrictions (e.g., licenses, fees) associated with any of the external resources that might apply to a future user? Please provide descriptions of all external resources and any restrictions associated with them, as well as links or other access points, as appropriate.
The dataset is self-contained, with no reliance on external or dynamic resources.

**Does the dataset contain data that might be considered confidential (e.g., data that is protected by legal privilege or by doctor-patient confidentiality, data that includes the content of individuals' non-public communications)?** If so, please provide a description.
No. We do not release any real user queries that we collected for fine-tuning and data synthesis.

**Does the dataset contain data that, if viewed directly, might be offensive, insulting, threatening, or might otherwise cause anxiety?** If so, please describe why.

No.

**Does the dataset relate to people?**  If not, you may skip the remaining questions in this section.
No.

**Does the dataset identify any subpopulations (e.g., by age, gender)?**  If so, please describe how these subpopulations are identified and provide a description of their respective distributions within the dataset.
No.

**Is it possible to identify individuals (i.e., one or more natural persons), either directly or indirectly (i.e., in combination with other data) from the dataset?**  If so, please describe how.
No.

**Does the dataset contain data that might be considered sensitive in any way (e.g., data that reveals racial or ethnic origins, sexual orientations, religious beliefs, political opinions or union memberships, or locations; financial or health data; biometric or genetic data; forms of government identification, such as social security numbers; criminal history)?**  If so, please provide a description.
No.

**Any other comments?**
The dataset's richness in diversity makes it a significant resource for advancing formal mathematics via AI.

### P.3  COLLECTION

**How was the data associated with each instance acquired?**  Was the data directly observable (e.g., raw text, movie ratings), reported by subjects (e.g., survey responses), or indirectly inferred/derived from other data (e.g., part-of-speech tags, model-based guesses for age or language)? If data was reported by subjects or indirectly inferred/derived from other data, was the data validated/verified? If so, please describe how.
The data was primarily derived from existing Lean 4 formal statements and proofs in mathlib and repositories. From these Lean statements, additional modalities were generated: informalizations, synthetic user queries and augmented proof states were produced by prompting LLMs.

**Over what timeframe was the data collected?**  Does this timeframe match the creation timeframe of the data associated with the instances (e.g., recent crawl of old news articles)? If not, please describe the timeframe in which the data associated with the instances was created. Finally, list when the dataset was first published.
All the data is collected in 2025.

**What mechanisms or procedures were used to collect the data (e.g., hardware apparatus or sensor, manual human curation, software program, software API)?**  How were these mechanisms or procedures validated?
We extract Lean statements from mathlib and other Lean repositories, and use GPT-4o API to generate the queries.

**What was the resource cost of collecting the data?**  (e.g. what were the required computational resources, and the associated financial costs, and energy consumption - estimate the carbon footprint.)
The total cost for GPT-4o API is around $2000.

**If the dataset is a sample from a larger set, what was the sampling strategy (e.g., deterministic, probabilistic with specific sampling probabilities)?**
N/A

**Who was involved in the data collection process (e.g., students, crowdworkers, contractors) and how were they compensated (e.g., how much were crowdworkers paid)?**
Graduate students and researchers with expertise in formal mathematics and AI.

**Were any ethical review processes conducted (e.g., by an institutional review board)?** If so, please provide a description of these review processes, including the outcomes, as well as a link or other access point to any supporting documentation.
No.

**Does the dataset relate to people?** If not, you may skip the remainder of the questions in this section.
No.

**Did you collect the data from the individuals in question directly, or obtain it via third parties or other sources (e.g., websites)?**
The Lean statements from our data was collected through mathlib and other Lean related github repositories.

**Were the individuals in question notified about the data collection?** If so, please describe (or show with screenshots or other information) how notice was provided, and provide a link or other access point to, or otherwise reproduce, the exact language of the notification itself.
N/A

**Did the individuals in question consent to the collection and use of their data?** If so, please describe (or show with screenshots or other information) how consent was requested and provided, and provide a link or other access point to, or otherwise reproduce, the exact language to which the individuals consented.
N/A

**If consent was obtained, were the consenting individuals provided with a mechanism to revoke their consent in the future or for certain uses?** If so, please provide a description, as well as a link or other access point to the mechanism (if appropriate)
N/A

**Has an analysis of the potential impact of the dataset and its use on data subjects (e.g., a data protection impact analysis)been conducted?** If so, please provide a description of this analysis, including the outcomes, as well as a link or other access point to any supporting documentation.
No.

**Any other comments?**
N/A

P.4    PREPROCESSING / CLEANING / LABELING

**Was any preprocessing/cleaning/labeling of the data done(e.g.,discretization or bucketing, tokenization, part-of-speech tagging, SIFT feature extraction, removal of instances, processing of missing values)?** If so, please provide a description. If not, you may skip the remainder of the questions in this section.
Yes. Formal Lean statements were extracted from mathlib and related repositories and filtering was

applied to keep only the relevant Lean statements. Filtering was also applied to clusters before generating synthetic user queries to avoid invalid synthetic user queries.

**Was the "raw" data saved in addition to the preprocessed/cleaned/labeled data (e.g., to support unanticipated future uses)?** If so, please provide a link or other access point to the "raw" data.
Yes, raw data and intermediate representations are retained for reproducibility and future use.

**Is the software used to preprocess/clean/label the instances available?** If so, please provide a link or other access point.
The tools and scripts for preprocessing will be released upon acceptance.

**Any other comments?**
No.

### P.5 USES

**Has the dataset been used for any tasks already?** If so, please provide a description.
Yes, it was used to train and evaluate the Lean Finder and benchmark its performance against other retrieval models.

**Is there a repository that links to any or all papers or systems that use the dataset?** If so, please provide a link or other access point.
N/A

**What (other) tasks could the dataset be used for?**
Beyond semantic search and retrieval, the dataset could support tasks such as query generation, informal-to-formal translation, retrieval-augmented theorem proving, and exploring human–AI interaction in formal reasoning systems.

**Is there anything about the composition of the dataset or the way it was collected and preprocessed/cleaned/labeled that might impact future uses?** For example, is there anything that a future user might need to know to avoid uses that could result in unfair treatment of individuals or groups (e.g., stereotyping, quality of service issues) or other undesirable harms (e.g., financial harms, legal risks) If so, please provide a description. Is there anything a future user could do to mitigate these undesirable harms?
N/A

**Are there tasks for which the dataset should not be used?** If so, please provide a description.
The dataset is not suitable for tasks unrelated to Lean 4.

**Any other comments?**
No.

### P.6 DISTRIBUTION

**Will the dataset be distributed to third parties outside of the entity (e.g., company, institution, organization) on behalf of which the dataset was created?** If so, please provide a description.
Yes, the dataset will be made publicly available for research purposes.

**How will the dataset will be distributed (e.g., tarball on website, API, GitHub)?** Does the dataset have a digital object identifier (DOI)?
The dataset will be distributed via GitHub and academic repositories, with accompanying

documentation.

**When will the dataset be distributed?**
The dataset is expected to be released following the ICLR 2026 conference.

**Will the dataset be distributed under a copyright or other intellectual property (IP) license, and/or under applicable terms of use (ToU)?** If so, please describe this license and/or ToU, and provide a link or other access point to, or otherwise reproduce, any relevant licensing terms or ToU, as well as any fees associated with these restrictions.
Yes, it will be distributed under a permissive license (e.g., CC BY-SA 4.0) to encourage research use.

**Have any third parties imposed IP-based or other restrictions on the data associated with the instances?** If so, please describe these restrictions, and provide a link or other access point to, or otherwise reproduce, any relevant licensing terms, as well as any fees associated with these restrictions.
All charts are subjected to their respective copyrights by the authors of this paper.

**Do any export controls or other regulatory restrictions apply to the dataset or to individual instances?** If so, please describe these restrictions, and provide a link or other access point to, or otherwise reproduce, any supporting documentation.
N/A

**Any other comments?**
Distribution will include detailed usage guidelines to ensure proper application of the dataset.

P.7    MAINTENANCE

**Who is supporting/hosting/maintaining the dataset?**
The authors of Lean Finder.

**How can the owner/curator/manager of the dataset be contacted (e.g., email address)?**
Contact information will be provided with the dataset release.

**Is there an erratum?** If so, please provide a link or other access point.
N/A

**Will the dataset be updated (e.g., to correct labeling errors, add new instances, delete instances)?** If so, please describe how often, by whom, and how updates will be communicated to users (e.g., mailing list, GitHub)?
Yes, we will frequently update the dataset to include new Lean statements in mathlib and other Lean related repositories we used.

**If the dataset relates to people, are there applicable limits on the retention of the data associated with the instances (e.g., were individuals in question told that their data would be retained for a fixed period of time and then deleted)?** If so, please describe these limits and explain how they will be enforced.
N/A

**Will older versions of the dataset continue to be supported/hosted/maintained?** If so, please describe how. If not, please describe how its obsolescence will be communicated to users.
Yes, previous versions will remain accessible for reproducibility.

**If others want to extend/augment/build on/contribute to the dataset, is there a mechanism for them to do so?** If so, please provide a description. Will these contributions be validated/verified? If so, please describe how. If not, why not? Is there a process for communicating/distributing these contributions to other users? If so, please provide a description.

Yes, contributions will be encouraged through a collaborative platform (e.g., GitHub).

**Any other comments?**

The dataset's maintainers are committed to ensuring its long-term usability and relevance for scientific research.

## Q  MISC.

**URL to our web service, model, and benchmark.** We will continually update our system to support more Lean projects and a growing database of Lean statements.

**Author statement & license information.** We the authors bear all responsibility in case of violation of rights.

**Dataset Structure.** All files are stored in the JSONL format. Each input modality will be stored in a separate JSONL file. Each sample will contain the query and the ground truth Lean statements. We will also release our corpus of Lean statements.

