# OpenReview forum: "Lean Finder: Semantic Search for Mathlib That Understands User Intents"
_ICLR.cc/2026/Conference — ICLR 2026 Poster_

### Official Review · Reviewer_XmGe · 2025-10-29

**Soundness:** 3
**Presentation:** 3
**Contribution:** 3
**Rating:** 6
**Confidence:** 3

**Summary:**

The paper introduces Lean Finder, a semantic search engine for Lean and mathlib designed to better reflect the intents of mathematicians. It aims to improve retrieval of relevant theorems by moving beyond naive string or informalized statement matching. The system clusters semantic patterns from public Lean discussions, fine-tunes text embeddings on synthesized user-style queries, and incorporates human preference feedback to align results with mathematicians’ goals. Evaluations show roughly 30% relative improvement over existing Lean search engines and GPT-4o on several retrieval benchmarks. The method integrates cleanly with LLM-based theorem provers, offering a bridge between retrieval and formal reasoning.

**Strengths:**

1. Timely and useful application: it addresses a real pain point in the Lean community (retrieving theorems efficiently), motivation is clear, relevant, and meaningful.
2. Reasonable methodology grounding and convincing empirical validation: The pipeline is technically consistent and design decisions are reasonably justified, even if not groundbreaking. The quantitative results (e.g., relative gain) demonstrate significant improvement over SOTA baselines.
3. Writing is coherent, the structure clear, and the contribution easy to follow despite imperfect figures. And also, they released their code already. This is very helpful in understanding more details of their work.

**Weaknesses:**

1. Significant improvement can be made to the figures, especially Fig 1 and Fig 5, they are not even close to informative. It would be better off without them if your figure content can be replaced by a sentence or two.
2. Not so much discussion on the results. Would like to see more investigation into what makes your method so different from everyone else's and what makes the improvement so significant, if possible (maybe through some solid ablation study, if pages permit).
3. Real user feedback is integrated in the simplest form possible, basically just upvoting and downvoting. This is a good starting point though, but still kind of limited.

**Questions:**

1. Can the pipeline be extended to include more types of interactions with human feedback involved?
2. Is it possible to provide ablation studies on each component to evaluate more on their contributions?
3. I have some concern on the use of OpenAI API since they are constantly updated and their code/weight is not released to the public, and I would personally doubt if relying on it will harm the reproducibility of your results. How would you justify this part?

---

> ### Author Response · Authors · 2025-11-20
>
> We truly thank the time and effort of Reviewer XmGe in reviewing our paper, and for highlighting the useful application of our work to the community and the solid methodological grounding and convincing empirical validation.
>
> > **Q1.** Figures in paper can be improved to be more informative, especially Figure 1 and 5.
>
> Thanks for the suggestion! We promise to make our teaser figure (Figure 1) and the training pipeline figure  (Figure 5) clearer and more informative by connecting them more to the subsections in methods and experiments.
>
> > **Q2.** Would like to see investigation into what makes your method improve so significant via ablation studies
>
> **We have conducted ablation studies** where Lean Finder was trained without **synthetic user queries** and without the **DPO fine-tuning stage**, keeping training steps and other hyperparameters fixed (Appendix J: Tables 10 and 11). **Removing either component leads to a clear drop in real-user preference**, demonstrating that our user-intent–driven synthetic query generation and DPO-based preference alignment are key to Lean Finder’s gains over baselines.
>
> > **Q3.** Can you extend the pipeline to include more types of human feedback?
>
> Thanks for the great advice! Our pipeline is flexible and can incorporate more diverse user feedback during RLHF beyond the retrieval and model level feedback we already use. We plan to extend this with additional signals, such as ranking-based feedback.
>
> > **Q4.** Using OpenAI API might hurt reproducibility of the results.
>
> We promise to release the entire dataset, and we have already released the prompt and the OpenAI API configuration in the paper (Appendix C, Appendix N). This can be used to generate more data if needed.

---

> ### Author Response · Authors · 2025-11-26
> **Looking forward to discussions**
>
> Dear Reviewer XmGe:
>
> As the author-reviewer discussion period has started for several days now and will end soon, we will appreciate if you could check our response to your review comments soon. This way, if you have further questions and comments, we can still reply before the author-reviewer discussion period ends. If our response resolves your concerns, we kindly ask you to consider raising the rating of our work. Thank you very much for your time and efforts.
>
> Best regards,
>
> Authors of Submission #12884

---

### Official Review · Reviewer_2WYg · 2025-10-29

**Soundness:** 2
**Presentation:** 3
**Contribution:** 2
**Rating:** 4
**Confidence:** 4

**Summary:**

This paper proposes a semantic search engine for Lean and its mathematical library mathlib, named Lean Finder. The core objective of this work is to better understand the true intent of mathematicians to address the inefficiency of existing search tools when handling ambiguous, non-canonical queries. The authors mine user intent by analyzing public Lean community discussions and, based on this, "reverse-engineer" a large-scale training dataset of simulated real-world user queries. The model is fine-tuned using contrastive learning and Direct Preference Optimization (DPO) to align with user preferences. Experimental results show that Lean Finder achieves significant improvements over the existing Lean Search and the general-purpose GPT-4o model across multiple evaluation metrics, and it received a higher preference rate in a user study.

**Strengths:**

1.  **High Importance of the Problem:** How to efficiently find required theorems in the vast mathlib library is a recognized pain point for the Lean community and the entire field of formal mathematics. This paper is dedicated to solving a very practical and impactful problem.

2.  **Potential Community Value:** If the performance improvements reported in the paper hold up under fairer comparisons, Lean Finder could become a very useful tool for Lean developers and mathematicians, potentially increasing the efficiency of formalization work significantly.

**Weaknesses:**

Although the motivation and basic idea of this paper are commendable, there are several major flaws in demonstrating its effectiveness, which severely impact the credibility of its conclusions.

1.  **Baselines for Comparison are Severely Insufficient and Outdated:**
    *   The paper's primary comparison is with Lean Search (Gao et al., 2024a/b). Some time has passed since that work was published. In the interim, other new search tools have emerged in the Lean community, such as **LeanExplore (Asher, 2025)**. These tools may use different technical approaches or index a broader range of data. A paper claiming to achieve state-of-the-art (SOTA) performance should be comprehensively compared with all relevant systems, especially the latest ones. Merely outperforming a system from over a year ago is not sufficient to prove its superiority.

2.  **Obvious Fairness Issues in Experimental Setup and Comparison Methods:**
    *   **The comparison with Lean Search is likely unfair:** The paper repeatedly emphasizes its advantage in handling "ambiguous user queries." However, the original Lean Search paper and system include a key technique specifically designed for such queries: **query augmentation**. This paper does not seem to have enabled or discussed this feature in its comparative experiments, instead directly testing the system with raw, ambiguous queries, which it may not be optimized for. This constitutes a comparison under an "unintended use case," and the results naturally favor the proposed model. A fair comparison should be conducted with both systems running in their optimal configurations.
    *   **Using GPT-4o as a primary baseline is not convincing:** GPT-4o is a powerful general-purpose large language model, but it is not a specialized model specifically optimized for the highly vertical task of Lean theorem retrieval. It may not be familiar with Lean 4 syntax, the internal structure of mathlib, or its naming conventions. Therefore, it is not surprising that Lean Finder surpasses GPT-4o. More importantly, this does not prove that it is superior to other **purpose-built** theorem search systems.
    *   **The numerical results are consequently less persuasive:** The paper's claimed "over 30% relative improvement" was obtained on the aforementioned unfair baselines. If compared against Lean Search with query augmentation enabled or other more advanced systems, this advantage might significantly shrink or even disappear.

3.  **Limited Methodological Innovation:**
    *   The core techniques of this paper—**Contrastive Learning** for training retrieval models, synthesizing data oriented towards user intent, and **Direct Preference Optimization (DPO)** for aligning with human feedback—are all mature and standard techniques in the current fields of retrieval and alignment. While successfully applying these techniques to the Lean domain is a valuable engineering practice, from the perspective of algorithmic innovation, this paper does not propose a new model architecture or training paradigm.

**Questions:**

1.  Regarding the choice of baselines, why did you not include newer Lean search systems like LeanExplore in your comparison? Given the existence of these systems, how do you support Lean Finder's leading position?
2.  When comparing with Lean Search, did you consider its "query augmentation" feature?

---

> ### Author Response · Authors · 2025-11-20
>
> We truly thank the time and effort of reviewer 2WYg in reviewing our paper, and highlighting the significance of the problem our paper attempts to solve, as well as the potential community value our work offers.
>
> > **Q1.** No comparison with Lean Explore
>
> After ensuring the test queries’ ground truth Lean code appear in Lean Explore’s static database, we show results comparing Lean Finder and Lean Explore on all input types that Lean Explore supports:
>
> | Model                          | Input Type                      | R@1  | R@5  | R@10 | MRR  |
> |:-------------------------------|:-----------------------------|:----:|:----:|:----:|:----:|
> | Lean Finder (ours)            | Informalized statement       | **65.7** | **89.0** | **93.2** | **0.76** |
> | Lean Explore                  | Informalized statement       | 35.0 | 61.8 | 68.0 | 0.46 |
> | |                   |      |      |      |      |
> | Lean Finder (ours)            | Synthetic user query         | **57.3** | **85.4** | **91.2** | **0.69** |
> | Lean Explore                  | Synthetic user query         | 26.3 | 51.3 | 60.1 | 0.37 |
> | |                              |      |      |      |      |
> | Lean Finder (ours)            | Augmented statement          | **86.8** | **96.7** | **98.0** | **0.91** |
> | Lean Explore                  | Augmented statement          | 85.8 | 93.6 | 95.2 | 0.89 |
>
> **Lean Finder still significantly outperforms Lean Explore**, especially on natural language queries like informalized statements and user queries. This shows Lean Finder’s strengths and advantages still holds when compared with the newest tool like Lean Explore.

---

> ### Author Response · Authors · 2025-11-20
>
> > **Q2.** No comparison with Lean Search using query augmentation mode
>
> We present the results of Lean Search under two modes: **augmentation mode and normal mode (the mode used in our paper)**. In augmentation mode, the input query is first augmented using Lean Search’s built-in query augmentation functionality, and the augmented version is then submitted to Lean Search.
>
> | Model                          | Input Type                      | R@1  | R@5  | R@10 | MRR  |
> |:-------------------------------|:-----------------------------|:----:|:----:|:----:|:----:|
> | Lean Finder (ours)            | Informalized statement       | **64.2** | **88.9** | **93.3** | **0.75** |
> | Lean Search (augmentation mode)                  | Informalized statement       | 31.9 | 77.4 | 84.8 | 0.44 |
> | Lean Search (normal mode)                  | Informalized statement       | 49.2 | 76.5 | 82.5 | 0.61 |
> | |                   |      |      |      |      |
> | Lean Finder (ours)            | Synthetic user query         | **54.4** | **84.4** | **91.4** | **0.68** |
> | Lean Search (augmentation mode)                   | Synthetic user query         | 30.0 | 72.7 | 83.3 | 0.43 |
> | Lean Search (normal mode)                   | Synthetic user query         | 47.1 | 77.7 | 83.7 | 0.60 |
> | |                              |      |      |      |      |
> | Lean Finder (ours)            | Augmented statement          | **82.7** | **97.0** | **97.7** | **0.89** |
> | Lean Search (augmentation mode)                   | Augmented statement          | 45.4 | 86.0 | 93.0 | 0.58 |
> | Lean Search (normal mode)                   | Augmented statement          | 59.2 | 81.9 | 85.5 | 0.69 |
>
> As shown above, **Lean Search R@1 decreases with query augmentation, while R@5 and R@10 stay comparable or slightly improve relative to the non-augmented setting**. We further note that Lean Search’s underlying database seems to have been updated to a newer Mathlib version, since several Lean statements now appear under different paths than when our initial experiments were conducted.
>
> We now **provide a case study** to better understand the performance difference of Lean Search with query augmentation mode:
>
> >>**The original query**:
>
> When working with functors being isomorphic in category theory, how can I manage natural transformations in Lean to show properties like being homological transfer from one functor to another?
>
> >>**The augmented query**:
>
> Natural Transformation Management in Lean for Functor Isomorphism: When dealing with two functors that are isomorphic in category theory, one can utilize natural transformations within the Lean proof assistant to demonstrate that certain categorical properties transfer between these functors. Specifically, if you have an isomorphism between functors F and G, you can construct natural transformations that map between their respective compositions with other functors, allowing you to prove that properties like exactness in homological algebra, preservation of limits or colimits, or other categorical structures are preserved under this isomorphism. This involves defining appropriate naturality conditions and verifying that diagrams commute through careful application of Lean's category theory library, ensuring that all components properly interact to maintain the desired structural relationships between the functors.
>
> >>**Ground truth code**:
> theorem IsHomological.of_iso {F₁ F₂ : C ⥤ A} [F₁.IsHomological] (e : F₁ ≅ F₂) : F₂.IsHomological :=
>   have := preservesZeroMorphisms_of_iso e
>   ⟨fun T hT => ShortComplex.exact_of_iso (ShortComplex.mapNatIso _ e)
>     (F₁.map_distinguished_exact T hT)⟩
>
> With the original query, this theorem is ranked 1st by Lean Search (and ranked 1st by Lean Finder); with the augmented query, it appears at rank 19 by Lean Search (and ranked 3rd by Lean Finder).
>
> Intuitively, **augmented query mentions additional topics** (natural transformations, exactness, limits, etc.). This richer context could help Lean Search retrieve more relevant results which can explain the improved R@5/R@10. Yet, this may also **increase query size, inject noise, and shift the focus away from the user’s actual intent** (transferring “homological” between isomorphic functors) which hurts sensitive metrics such as R@1.
>
> **We admire the positive effect of query augmentation in Lean Search** and its ability to surface relevant Lean statements. Our Lean Finder takes a complementary approach: instead of expanding queries at inference time, we align the model to the expected query types during training, using user queries and intent clustering. This design helps Lean Finder remain robust to diverse, natural user queries while keeping the best matches for user’s intent near the top of the ranking. This is demonstrated by the stronger performance of Lean Finder compared to other systems.

---

> > ### Comment · Reviewer_2WYg · 2025-11-23
> >
> > These results are truly outstanding. Based on the data you’ve presented, I’m confident that Lean Finder will have a substantial impact on the community. Would it be possible for you to share part of your original benchmark dataset so I can test on them myself? I would be very willing to significantly raise your score afterward.

---

> > > ### Author Response · Authors · 2025-11-23
> > >
> > > We thank reviewer 2WYg for reading our response and providing the reply.
> > >
> > > As requested, we have included the data in the Supplementary Material (see the formatted_data directory after unzipping). We sincerely appreciate your engagement and your willingness of raising the score of our work.

---

> > > > ### Comment · Reviewer_2WYg · 2025-11-24
> > > >
> > > > Sorry for overlooking the material in the supplementary files earlier, and thank you for pointing it out! I’ve noticed that the query style in your dataset is rather unusual. As far as I know, most users of theorem search engines typically do not write queries in this style—real queries are usually short and descriptive. The queries in your benchmark seem to be largely based on a particular method of synthetic generation, and very likely share the same style as your training data. This kind of systematic bias weakens the persuasiveness of the “high performance” you claim.
> > > >
> > > > In addition, based on my observations, a single query in your benchmark can correspond to multiple valid answers, yet you provide only one ground truth. This design does not quite align with the expectations for a robust benchmark.
> > > >
> > > > Could you clarify these issues? Thank you again!

---

> > > > > ### Author Response · Authors · 2025-11-24
> > > > >
> > > > > We thank reviewer 2WYg for reading our response and providing this further reply! We address your questions below:
> > > > >
> > > > > > **Q1. potential mismatch in length and style between synthetic queries and real user queries**
> > > > >
> > > > > We provide three evidence to validate the quality, realism, and user-alignment of our dataset:
> > > > >
> > > > > 1. Our synthetic user query length is similar to the length of the informalizations from Herald dataset[1], which Lean Search relies on. Despite similar lengths, Lean Finder outperforms Lean Search. **This indicates that token length is not the determining factor for performance**. In addition, while our synthetic queries have similar length as Herald informalizations, the **PCA visualization of embeddings in Figure 2 demonstrates that our synthetic user queries align significantly better with real user queries than the Herald informalizations do**.
> > > > >
> > > > > 2. We further clarify the nature of real world queries based on our recent deployment logs. The real user queries have average token count (with gpt-4o tokenizer) of 20 tokens, while our synthetic user queries have average token count of 34 tokens. While synthetic queries are slightly more verbose, our logs confirm that **real users predominantly use sentence-like, natural language structures, mirroring the linguistic patterns of our synthetic data**, rather than extremely short, keyword-only style.
> > > > >
> > > > > 3. **Our ablation studies further validates the quality of the synthetic user queries**. We found that removing synthetic user queries from training significantly decreased the model's preference rate among real human evaluators (Table 13). If there were a severe distribution mismatch or "unusual" style bias, training on this data would harm performance on real tasks. Instead, **the improved human preference confirms that our synthetic queries accurately model the distribution of real user queries and intents and generalize effectively to real-world scenarios**.
> > > > >
> > > > >
> > > > > > **Q2. Concern regarding potential stylistic overlap between training and evaluation data.**
> > > > >
> > > > > 1. To address this concern, **we conducted user study (Section 4.2), using real user Lean/mathlib queries sourced from GitHub (not directly sharing the same style as our training data)**. The results (Table 3) show that participants ranked Lean Finder’s retrieval results as the "Most Helpful" **nearly twice as often** as the strongest baselines. Since this evaluation utilized purely human-written queries, the "systematic bias" argument cannot apply here. **The strong human preference confirms that our model’s performance translates directly to real-world usage.**
> > > > >
> > > > > 2. Also, for the Informalized Statement input type in our test set, we utilized a different informalization pipeline (distinct prompts) compared to the training data generation to ensure fairness in comparison (Appendix G). **Despite this shift in informalization style, Lean Finder still significantly outperforms baselines**. This demonstrates that Lean Finder is not memorizing a specific "synthetic style", but has **robustly learned the underlying semantic alignment between natural language and formal mathematics.**
> > > > >
> > > > > > **Q3. A single query can correspond to multiple valid answers**
> > > > >
> > > > > We address this concern by distinguishing between two high level input types in our dataset.
> > > > >
> > > > > 1. For the **proof state input type**, it is true that a single proof state query can correspond to multiple ground truth Lean statements (multiple theorems can be used in a single tactic), and we indeed **have fully accounted for multiple ground-truth in our dataset and during evaluation for all baselines**. The full dataset that will be released will explicitly show this.
> > > > >
> > > > > 2. For **natural language queries**, they are designed to have one Lean statement that best answers and addresses the query (eg. each synthetic query is reversely synthesized based on a single Lean statement). **This is consistent with other informalization dataset like Herald[1] and MMA[2] which also assumes a single ground truth for each natural language query**. We acknowledge that natural language may be inherently ambiguous and other theorems may be related or partially applicable. To strictly validate our performance despite the inherent ambiguity in natural queries, we conducted user study (Section 4.2) where **experienced Lean users evaluated the retrieval results based on actual utility, not just a rigid match against a single ground truth**. Lean Finder was preferred significantly more often than the baselines, and this confirms that Lean Finder’s strong performance in the benchmark **reflects a genuine ability to retrieve the most relevant and helpful theorems for users.**
> > > > >
> > > > > [1] “Herald: A Natural Language Annotated Lean 4 Dataset” Gao et al. 2024
> > > > >
> > > > > [2] “Multilingual Mathematical Autoformalization” Jiang et al. 2023
> > > > >
> > > > >
> > > > >
> > > > > We hope this response resolves your further questions. We are happy to discuss further!

---

> > > > > > ### Comment · Reviewer_2WYg · 2025-11-26
> > > > > >
> > > > > > Regarding Q1, your arguments are sufficient to convince me that your synthetic data can serve as a reasonably high-quality training dataset. However, as an evaluation benchmark, being merely “close to real user queries” is not persuasive enough—especially since the degree of similarity itself is not quite adequate. The average token length differs significantly, and I still find it hard to believe that queries like “How can I ensure Lean correctly finds or constructs a `MulActionHomClass` instance when dealing with transformation mappings between two spaces with monoidal actions?” or “Is there a theorem in Lean 4 that expresses the relationship between the span of a union of sets and the supremum of their spans? I'm particularly interested in scenarios where each set is itself indexed by another index set.” truly match the style of queries used by theorem-search engine users. This does not align with my own experience or my observations of others.
> > > > > >
> > > > > > Regarding Q2, could you clarify what exactly “real user Lean/mathlib queries sourced from GitHub” refers to?
> > > > > >
> > > > > > Regarding Q3, the autoformalization task is largely unrelated to retrieval. In retrieval tasks, queries tend to be much vaguer. Still, your analysis in Section 4.2 is indeed convincing.

---

> > > > > > > ### Author Response · Authors · 2025-11-27
> > > > > > > **Thank you for your reply!**
> > > > > > >
> > > > > > > Thank you for your reply. We address your questions and concerns below:
> > > > > > >
> > > > > > > >Q1. Concern regarding the misalignment between synthetic queries and real-world user queries in the evaluation.
> > > > > > >
> > > > > > > Firstly, **we thank you for acknowledging that the synthetic queries serve as high-quality training data to close the gap between user queries and informalization in search engines that previous work have ignored**.
> > > > > > >
> > > > > > > To directly address your concern and **demonstrate Lean Finder’s robustness to very short, keyword-based inputs**, we conducted an **additional evaluation using the Lean Explore dataset.**
> > > > > > >
> > > > > > > This dataset consists of 300 **short, keyword-based queries** (e.g., "Borel sets sigma-algebras" "Cantor's theorem uncountability reals"), **comprised mostly of 2–5 words, which aligns with the query style you described**. Using the similar evaluation settings as Lean Explore (retrieving 5 statements from each model, and using GPT-4o for evaluation), we show the following results:
> > > > > > >
> > > > > > > | Model                 | 1st place rate (%)  | 2nd place rate (%)  | 3rd place rate (%) |
> > > > > > > |:-------------------------------:|:----:|:----:|:----:|
> > > > > > > | Lean Finder (ours)          | **49.0%** | 19.3% | 31.7%|
> > > > > > > | Lean Search                   | 38.7% | 38.3% | 23.0% |
> > > > > > > | Lean Explore  		   | 12.7% | 42.3% | 45.0% |
> > > > > > >
> > > > > > > **Lean Finder still outperforms the baselines even on this distinct query style**. The percentage of times Lean Finder is ranked as most helpful (1st place) is higher than both Lean Search and Lean Explore. This confirms that Lean Finder provides high-quality retrieval not just for natural language questions, but also for the concise, keyword-driven queries typical of theorem-search users.
> > > > > > >
> > > > > > > **The consistently strong performance of Lean Finder across diverse input styles and lengths demonstrates its robustness. As also noted by other reviewers, this confirms that Lean Finder provides great value to the community.**
> > > > > > >
> > > > > > > >Q2. Clarification on real user Lean/mathlib queries sourced from GitHub
> > > > > > >
> > > > > > > Thank you for the question. Due to ICLR’s policy, we must keep our responses fully anonymous and cannot release links to GitHub repositories before camera ready, but we have consulted with authors of these GitHub repositories and they agree to release their query text. The queries are now in supplementary material (See real_user_queries.txt after unzip).
> > > > > > >
> > > > > > > >Q3. The autoformalization task is largely unrelated to retrieval, but your analysis in Section 4.2 is indeed convincing.
> > > > > > >
> > > > > > > Thank you for acknowledging the analysis is convincing! Although autoformalization is not directly related to retrieval, it shows that assuming a single ground truth Lean code match for each natural language description is widely used in Lean despite inherent ambiguity in natural language.
> > > > > > >
> > > > > > > We hope the responses have addressed your concerns. We are happy to discuss this further!

---

> > > > > ### Author Response · Authors · 2025-11-26
> > > > > **Looking forward to more discussions**
> > > > >
> > > > > Dear Reviewer 2WYg,
> > > > >
> > > > > As the author-reviewer discussion period is nearing its end, we would greatly appreciate it if you could review our responses to your comments at your earliest convenience.
> > > > >
> > > > > This will allow us to address any further questions or concerns you may have before the discussion period concludes.
> > > > >
> > > > > If our responses satisfactorily address your concerns, we kindly ask you to consider revising your rating of our work.
> > > > >
> > > > > Thank you very much for your time and efforts.
> > > > >
> > > > > Best regards,
> > > > >
> > > > > Authors of Submission #12884

---

> ### Author Response · Authors · 2025-11-20
>
> > **Q3.** Using GPT-4o as baseline is not convincing, and does not show Lean Finder’s superiority to other specialized Lean search systems.
>
> 1. **Lean Finder is already compared against and outperforms specialized Lean search engines** (Lean Search, and Lean Explore in our new experiments).
>
> 2. GPT-4o/ChatGPT is a widely used assistant for day-to-day and scientific work, including by mathematicians and researchers [1]. With search mode enabled, it can browse the web and access Lean/mathlib documentation and code. Our goal is not to treat GPT-4o as a specialized theorem retriever, but **to compare Lean Finder against the powerful general-purpose tool** that many practitioners already rely on, which makes it **a practically important and relevant baseline**.
>
> [1] “CHATGPT FOR SCIENTIFIC DISCOVERY: ENHANCING OR UNDERMINING INNOVATION?” Deckker et al. 2025
>
> > **Q4.** The numerical results are less persuasive since no comparison with Lean Explore and Lean Search with query augmentation mode enabled.
>
> The comparisons with both Lean Explore and Lean Search with query augmentation enabled that are presented above show that **Lean Finder still consistently outperforms these systems across input types**.
>
> > **Q5.** Limited methodological innovation on model architecture or training paradigm
>
> The core novelty of Lean Finder **is not about training techniques or model architectures**, and in fact, the idea of contrastive learning is widely used in code search papers [1, 2, 3]. Rather, **our novelty lies in pinpointing and solving the core and overlooked bottleneck of current Lean search engines which is to natively align the search engines with human intents**. This blind spot is never discussed, explored, or even realized in existing search engines papers in Lean. Our novel data synthesis pipeline **closes the domain gap between real user queries and mechanical informalization of Lean code**, and as pointed out by Reviewer xHpC and jNAc, our work solves the problem of scarcity in high quality real user queries in Lean.
>
> The distribution gap between generated descriptions of code and real user queries has been examined in LLM-based code search for Python with inference-time techniques [4], but **it has never been studied in the AI4Math domain**.  Motivated by this observation, we came up with the novel “reverse engineering” method to synthesize user queries for training to solve the misalignment, as justified by embedding distance analysis shown in Figure 2 and the strong performance of Lean Finder in the user study. **As Reviewer xHpC comments: our method “Innovatively clusters real user intents (5 categories) to guide synthetic query generation, solving the scarcity of high-quality real user query data”** and releases “the largest Lean code search dataset to date, laying a solid foundation for model training.”
>
> [1] “CodeT5: Identifier-aware Unified Pre-trained Encoder-Decoder Models for Code Understanding and Generation” Wang et al. 2021
>
> [2] “CodeT5+: Open Code Large Language Models for Code Understanding and Generation” Wang et al. 2023
>
> [3] “CoCoSoDa: Effective Contrastive Learning for Code Search” Shi et al. 2023
>
> [4] “Enriching Query Semantics for Code Search with Reinforcement Learning” Wang et al. 2021

---

### Official Review · Reviewer_jNAc · 2025-10-29

**Soundness:** 3
**Presentation:** 2
**Contribution:** 3
**Rating:** 4
**Confidence:** 3

**Summary:**

This paper proposes an intelligent search tool for the Lean 4 formal mathematical proof system—Lean Finder. This system aims to help mathematical researchers and automated theorem proving models more efficiently retrieve lemmas and theorems relevant to their current proof tasks.

It analyzes queries posted by real mathematicians on public platforms and clusters them into different user intentions. Based on these user intentions, it generates synthetic user queries for Lean formal statements to build a training dataset.

Then, through contrastive learning and the DPO algorithm, the large model is fine-tuned to obtain a model that better matches human query intentions. The model achieves good performance.

**Strengths:**

1. Lean Finder addresses the scarcity of query and answer data in Lean when constructing datasets. It first clusters queries from real mathematicians to identify different user intents, then generates prompts based on these intents. Secondly, to construct data pairs, it utilizes a large model to synthesize inverse queries based on Lean formal statements, resulting in a large dataset.

2. During model training, it fine-tunes the model using contrastive learning and DPO methods, achieving better results compared to current methods.

**Weaknesses:**

1. While the paper describes how synthetic user queries are generated from formal Lean statements, it does not provide a quantitative or qualitative analysis of the semantic quality, correctness, or diversity of these synthesized queries. Such an evaluation would be crucial to demonstrate the reliability of the training data.

2. The fine-tuning experiments are conducted solely on the DeepSeek-Prover-V1.5-RL 7B model. It remains unclear whether the proposed approach generalizes to other base models or architectures. Moreover, the paper lacks a comparison with existing large retrieval models that could serve as relevant baselines.

3. In the experiment that examines whether Lean Finder can serve as a prover-agnostic retrieval tool, the paper does not include results combining other Lean search or Lean state search methods with the referenced provers.

**Questions:**

In line 201, the paper mentions “high-quality discussion.” Could the authors clarify how “high-quality” is defined in this context? Specifically, what criteria or metrics are used to determine whether a discussion qualifies as high-quality data?

---

> ### Author Response · Authors · 2025-11-20
>
> We truly thank the time and effort of reviewer jNAc in reviewing our paper, and highlighting our novel approach in synthesizing user queries to tackle the lack of real user data, and our rigorous training pipeline.
>
> > **Q1.** Need analysis of the semantic quality, correctness, or diversity of synthetic data.
>
> Thanks for bringing this up! This is a very important point, and we indeed value the quality of the synthetic data. We address this question from three perspectives:
>
> 1. We already conducted ablation studies where Lean Finder was trained without synthetic queries, keeping training steps and hyperparameters identical (Appendix J, Tables 10 and 11). **Removing synthetic queries leads to a clear drop in both real user preference and retrieval performance**. This provides **quantitative evidence that the synthetic queries are of high quality** and are semantically aligned with real user queries and significantly improve performance of Lean Finder.
>
> 2. We control the quality of synthetic queries by using GPT-4o to (i) analyze the given Lean statement and determine which query clusters are applicable (Section 3.1.2). (ii) then synthesize queries grounded in the given Lean statement and the intent clusters. This **avoids unnatural cases where a query cluster is irrelevant to a statement**. We additionally performed **manual inspection** of a subset of synthetic queries to verify their faithfulness to Lean statements and user intent.
>
> 3. For the diversity of synthetic queries, we showed the distribution of synthetic queries across intent clusters (Figure 4a), and it **mirrors real Zulip discussion distribution**. For example, metaprogramming vs. lemma search accounts for ~13 vs. ~300 messages/week observed on Zulip, matching 1% vs. 41% of our synthetic data.
>
> We promise to emphasize this discussion in camera-ready, and move this discussion to earlier pages.
>
> > **Q2.** How does the training pipeline generalize to other base models?
>
> Our data synthesis and training methods are **model-agnostic**. Due to compute and time constraints, we were unable to finish fine-tuning additional base models in a short period of time. This is consistent with prior work: Lean State Search trained one architecture, and Lean Search and Lean Explore performs no training.
>
> > **Q3..** No comparison with existing retrieval models
>
> The specialized search engine baselines like Lean Search and Lean Explore both use embedding models (E5mistral-7b and BAAI bge-baseen-v1.5) at their core, therefore we do have comparison with existing retrieval models. However, we **further compare with one of the state-of-the-art embedding model of comparable size Qwen3-Embedding-8B**, which is even larger than our model size by 1B, and show the results below:
>
> | Model                          | Input Type                      | R@1  | R@5  | R@10 | MRR  |
> |:-------------------------------|:-----------------------------|:----:|:----:|:----:|:----:|
> | Lean Finder (ours)            | Informalized statement       | **64.2** | **88.9** | **93.3** | **0.75** |
> | Qwen3-Embedding-8B                  | Informalized statement       | 57.7 | 82.5 | 89.2 | 0.69 |
> | |                   |      |      |      |      |
> | Lean Finder (ours)            | Synthetic user query         | **54.4** | **84.4** | **91.4** | **0.68** |
> | Qwen3-Embedding-8B                  | Synthetic user query         | 41.2 | 73.8 | 82.8 | 0.55 |
> | |                              |      |      |      |      |
> | Lean Finder (ours)            | Augmented statement          | **82.7** | **97.0** | **97.7** | **0.89** |
> | Qwen3-Embedding-8B                  | Augmented statement          | 76.2 | 93.4 | 95.1 | 0.83 |
> | |                              |      |      |      |      |
> | Lean Finder (ours)            |  Augmented proof state         | **24.6** | **56.8** | **67.9** | **0.40** |
> | Qwen3-Embedding-8B                  | Augmented proof state      | 8.0 | 26.1 | 34.7 | 0.17 |
> | |                              |      |      |      |      |
> | Lean Finder (ours)            |  Raw proof state         | **8.3** | **30.1** | **40.0** | **0.19** |
> | Qwen3-Embedding-8B                  | Raw proof state      | 5.4 | 21.1 | 27.4 | 0.13 |
>
> Our Lean Finder also **outperforms the state-of-the-art embedding model like Qwen3-Embedding-8B**, showing Lean Finder’s strong ability on all input types.
>
> Due to time limits and constraints on GPU resources, we can’t fine-tune this model on our dataset in a short period of time, but we promise to include this experiment in camera-ready.

---

> ### Author Response · Authors · 2025-11-20
>
> > **Q4.** No experiments that combine other retrievers with referenced provers.
>
> We integrate Lean State Search with the referenced provers using the same settings described in the paper. We exclude Lean Search because our integration method requires the search engine to support proof states as input, and Lean Search does not support this input type.
>
> | Model                          | Dataset | pass@1  | pass@32  | pass@128 |
> |:-------------------------------|:----:|:----:|:----:|:----:|
> | Goedel-Prover-SFT                                       | MiniF2F Test | 38.1% | 57.7% | 59.4% |
> | Goedel-Prover-SFT w/ Lean Finder (ours)               | MiniF2F Test | **38.5%** | **58.2%** | **59.8%** |
> | Goedel-Prover-SFT w/ Lean State Search     | MiniF2F Test | 36.9% | 56.2% | 59.4% |
> ||||||
> | Goedel-Prover-SFT                                       | ProofNet Test | 6.6% | 13.2% | 17.6% |
> | Goedel-Prover-SFT w/ Lean Finder (ours)               | ProofNet Test | **7.2%** | 14.4% | 17.6% |
> | Goedel-Prover-SFT w/ Lean State Search     | ProofNet Test | 6.1% | 14.4% | 17.6% |
> ||||||
> | Goedel-Prover-SFT                                       | PutnamBench | **3** | 6 | 7 |
> | Goedel-Prover-SFT w/ Lean Finder (ours)               | PutnamBench | 2 | 6 | **7**|
> | Goedel-Prover-SFT w/ Lean State Search     | PutnamBench | 2 | 6 | 6 |
>
>
> | Model                          | Dataset | pass@1  | pass@32  | pass@128 |
> |:-------------------------------|:----:|:----:|:----:|:----:|
> | DeepSeek-Prover-V1.5-RL                                       | MiniF2F Test | 34.4% | 50.0% | 51.6% |
> | DeepSeek-Prover-V1.5-RL w/ Lean Finder (ours)               | MiniF2F Test | **35.7%** | **50.0%** | 51.6% |
> | DeepSeek-Prover-V1.5-RL w/ Lean State Search     | MiniF2F Test | 32.2% | 49.2% | 51.6% |
> ||||||
> | DeepSeek-Prover-V1.5-RL                                        | ProofNet Test | 4.3% | 14.5% | 17.7% |
> | DeepSeek-Prover-V1.5-RL  w/ Lean Finder (ours)               | ProofNet Test | **6.5%** | **18.3%** | **19.4%** |
> | DeepSeek-Prover-V1.5-RL  w/ Lean State Search     | ProofNet Test | 4.3% | 16.1% | 18.3% |
> ||||||
> | DeepSeek-Prover-V1.5-RL                                        | PutnamBench | 4 | 7 | 7 |
> | DeepSeek-Prover-V1.5-RL    w/ Lean Finder (ours)               | PutnamBench | **5** | 6 | 7 |
> | DeepSeek-Prover-V1.5-RL    w/ Lean State Search     | PutnamBench | 2 | **7** | 7 |
>
>
> | Model                          | Dataset | pass@8x8 |
> |:-------------------------------|:----:|:----:|
> | REALProver                                       | MiniF2F Test | 52.0% |
> | REALProver   w/ Lean Finder (ours)               | MiniF2F Test | 52.0% |
> | REALProver   w/ Lean State Search     | MiniF2F Test | 50.0% |
> ||||||
> | REALProver                                       | ProofNet Test | **23.7%** |
> | REALProver  w/ Lean Finder (ours)               | ProofNet Test | 23.1% |
> | REALProver  w/ Lean State Search     | ProofNet Test | 22.0% |
> ||||||
> | REALProver                                        | FATE-M | **51.1%** |
> | REALProver    w/ Lean Finder (ours)               | FATE-M | 49.6% |
> | REALProver    w/ Lean State Search     | FATE-M | 48.9% |
>
> Overall, Lean State Search exhibits inconsistent (either improved or degraded) proof success rate when integrated with different provers, whereas **Lean Finder yields more consistent and larger gains**. That said, there is no significant performance difference between the two integration settings since proof performance is mainly bottlenecked by the prover model.
>
> Our **main objective is to build a strong and user-centered search engine that aligns with real user queries**, and can integrate seamlessly with provers while preserving stability and delivering measurable retrieval benefits.
>
> > **Q5.** Can you clarify how “high-quality discussion” is defined in line 201?
>
> By high-quality discussions, we refer to Zulip threads that satisfy the following filtering criteria:
>
> 1. We only consider discussions from the past two years (starting in 2023), so that the **content reflects relatively recent Lean and mathlib usage**.
> 2. We only consider threads where **at least two participants are involved**, which indicates that the question is being engaged with rather than being a single spam-like post.
> 3. We **discard discussions that contain more than three hyperlinks**, as excessive linking often makes the core query harder to interpret.
>
> These heuristics help us focus on discussions that are active and semantically clear enough to serve as reliable data in the development of Lean Finder.

---

> ### Author Response · Authors · 2025-11-26
> **Anticipating Further Discussions**
>
> Dear Reviewer jNAc:
>
> We would like to kindly remind you that the author-reviewer discussion period has started for several days now and will end soon.
>
> We would greatly appreciate it if you could review our responses to your initial comments at your earliest convenience.
>
> This will enable us to address any additional queries or feedback you might have before the discussion period ends.
>
> Should our responses sufficiently address your concerns, we respectfully request that you consider raising the rating of our work.
>
> Thank you very much for your attention, time, and efforts!
>
> Best regards,
>
> Authors of Submission #12884

---

### Official Review · Reviewer_xHpC · 2025-11-01

**Soundness:** 3
**Presentation:** 3
**Contribution:** 3
**Rating:** 4
**Confidence:** 4

**Summary:**

This paper proposes **Lean Finder**, a user-centered semantic search engine for Lean and mathlib, aiming to address the bottlenecks of theorem localization difficulty and steep Lean 4 learning curve in formal theorem proving. The core approach involves: collecting 693 real user queries from Lean Zulip and GitHub, filtering and paraphrasing them with GPT-4o, clustering them into 5 user intent categories via OpenAI o3, generating synthetic user queries based on formal Lean statements and the 5 intents to build a 1.4M+ query-code pair dataset, and training a two-stage model (contrastive learning + DPO preference alignment). Evaluations show Lean Finder achieves over 30% relative improvement in retrieval performance compared to baselines like Lean Search and GPT-4o, with 81.6% user preference (from 5 participants) and compatibility with LLM-based provers. The authors promise to release code, models, and datasets upon acceptance, contributing to the Lean community.

**Strengths:**

1. **Targeted Problem-Solving**: Directly addresses core pain points in Lean usage (theorem localization and language learning curve) that hinder formal theorem proving progress, with strong practical relevance for mathematicians and Lean users.
2. **User-Centered Data Construction**: Innovatively clusters real user intents (5 categories) to guide synthetic query generation, solving the scarcity of high-quality real user query data. The 1.4M+ multimodal query-code dataset (synthetic queries, informalized statements, etc.) is the largest Lean code search dataset to date, laying a solid foundation for model training.
3. **Rigorous Model Design**: The two-stage training pipeline (contrastive learning for embedding alignment + DPO for human preference alignment) balances retrieval performance and user adaptability, and evaluations across multiple input modalities (informalized statements, proof states) demonstrate robust generalization.
4. **Community Value**: The commitment to release code, models, and datasets will provide valuable resources for the Lean and formal theorem proving communities, promoting follow-up research and practical applications.

**Weaknesses:**

1. **Potential Bias in Query Filtering/Paraphrasing**: The use of GPT-4o to filter and paraphrase real user discussions may introduce inconsistencies—GPT-4o’s interpretation of "questions answerable by Lean statements" and its paraphrasing style could deviate from the original user intent, affecting the authenticity of subsequent intent clustering and synthetic query generation.
2. **Incomplete Baseline Comparison**: Fails to compare with newer Lean retrievers (e.g., LEAN-EXPLORE), nor does it analyze the advantages/disadvantages of such state-of-the-art tools. This limits the paper’s ability to fully demonstrate Lean Finder’s competitiveness in the current research landscape.
3. **Limited Coverage of User Intents**: The 5 clustered user intent categories may not fully cover all real-world user intents (there is likely a "difference set" from the complete set of intents). This could lead to synthetic queries that miss niche or underrepresented user needs, restricting Lean Finder’s applicability to diverse scenarios.
4. **Small-Scale User Preference Study**: The 81.6% user preference rate is based on only 5 participants, which is a small sample size. This may result in biased or ungeneralizable conclusions about user acceptance, weakening the credibility of the user-centric design claim.
5. **Lack of Solutions for Lean Version Differences**: Lean’s different versions may lead to changes in theorem paths or syntax, but the paper does not propose solutions to handle such version-related inconsistencies. This could reduce Lean Finder’s practicality when applied to different Lean versions.

**Questions:**

1. During the filtering and paraphrasing of real user discussions with GPT-4o (Section 3.1.1), what validation measures were taken to ensure that the processed queries do not deviate from the original user intent? For example, was there a manual check of a subset of queries or a comparison with user feedback?
2. Why were newer retrievers like LEAN-EXPLORE not included in the baseline comparison? If there were practical constraints (e.g., unavailable code), could you analyze the potential differences in design principles between Lean Finder and these tools, and explain how Lean Finder might excel or fall short?
3. Regarding the 5 user intent categories, did you test whether they can cover the intents of newly collected user queries (beyond the initial 693)? If not, do you have plans to expand or dynamically adjust the intent categories to reduce the "difference set" with the complete intent set?
4. Given the small sample size (5 participants) in the user preference study, do you plan to conduct a larger-scale user study (e.g., involving more Lean practitioners from different backgrounds) to verify the generalizability of the 81.6% preference rate?
5. For differences in theorem paths or syntax across Lean versions, what technical routes do you consider to make Lean Finder compatible with multiple versions? For example, will you build a version-aware index or a mapping between theorem names across versions?

**Details Of Ethics Concerns:**

The author mentioned that they conducted web scraping on Lean Zulip and GitHub. Are there any copyright issues with this?

---

> ### Author Response · Authors · 2025-11-20
>
> We truly thank the time and effort of reviewer xHpC in reviewing our paper, and highlighting our community values and our novel approach in synthesizing user queries to tackle the lack of real user data.
>
> > **Q1.** Ethics concern on web scraping.
> 1. We thank the reviewer for raising this point. For collection from Lean Zulip, **we did not perform web scraping**. We **used the official Zulip API** (https://zulip.com/api/) on public channels only. We **anonymized the discussions** to ensure no user identities are present during processing. In addition, we **do not store any of the original discussions or user identities**, and only the paraphrased versions of queries are retained and used for clustering and model training. In addition, there is a public webpage that archives Lean Zulip: https://leanprover-community.github.io/archive/ discussions, which is likely included in the training data of general-purpose LLMs.
> 2. For GitHub, we restricted ourselves to public discussions and explicitly **contacted the users involved to obtain their consent for research use of their comments**; we can provide proof of this consent upon request. As stated in Appendix M, we promise to only release the synthesized content (plus open‑source Lean code under its existing licenses), and never raw Zulip text.
>
> > **Q2.** What validation measures were applied during query filtering from Zulip and paraphrasing to minimize bias?
>
> 1. We carefully designed the prompt to perform query filtering and paraphrasing, using **detailed instructions and few-shot examples** (Appendix N.2). We also performed **human inspection** to verify that its judgments match human intuition and that paraphrases preserve the original intent. Moreover, although GPT-4o’s interpretation of "questions answerable by Lean statements" may not be perfect, **our goal is to summarize user intents rather than to collect ground-truth Lean statements to the queries**. Even some queries retained are not answerable by Lean statements, they still expose meaningful user intent.
>
> 2. **Paraphrasing is necessary** because real Zulip queries contain discussion-forum noise (e.g., @-mentions, hyperlinks). **Paraphrasing protects privacy and removes discussion-forum noise**, and makes the final text better reflect what users would type into a search engine.
>
> > **Q3.** No Comparison with Lean Explore
>
> After ensuring the test queries’ ground truth Lean code appear in Lean Explore’s static database, we show results comparing Lean Finder and Lean Explore on all input types that Lean Explore supports:
>
> | Model                          | Input Type                      | R@1  | R@5  | R@10 | MRR  |
> |:-------------------------------|:-----------------------------|:----:|:----:|:----:|:----:|
> | Lean Finder (ours)            | Informalized statement       | **65.7** | **89.0** | **93.2** | **0.76** |
> | Lean Explore                  | Informalized statement       | 35.0 | 61.8 | 68.0 | 0.46 |
> | |                   |      |      |      |      |
> | Lean Finder (ours)            | Synthetic user query         | **57.3** | **85.4** | **91.2** | **0.69** |
> | Lean Explore                  | Synthetic user query         | 26.3 | 51.3 | 60.1 | 0.37 |
> | |                              |      |      |      |      |
> | Lean Finder (ours)            | Augmented statement          | **86.8** | **96.7** | **98.0** | **0.91** |
> | Lean Explore                  | Augmented statement          | 85.8 | 93.6 | 95.2 | 0.89 |
>
> **Lean Finder still significantly outperforms Lean Explore**, especially on natural language queries like informalized statements and user queries. This shows Lean Finder’s advantages still holds when compared with the newest tool like Lean Explore.
>
> > **Q4.** No analysis of differences with other Lean search tools
>
> Our advantage over existing search tools comes from our **novel large-scale synthetic queries that better reflect real user intent, closing the gap between real queries and mechanical informalizations of Lean code that previous work ignored**. This is evident from the embedding distance analysis in Figure 2 and Lean Finder’s strong user-study performance.

---

> ### Author Response · Authors · 2025-11-20
>
> > **Q5.** Limited coverage of user intents
>
> Our 5 intent categories are derived from 693 filtered real-world queries **sourced from the highly active  channels on Zulip with rich retrieval-related user intents** (e.g., #lean4, #mathlib4, #Is there code for X, #metaprogramming/tactics, etc.) and GitHub. We prioritized these active sources to ensure **our method captures the dominant search modes that constitute the vast majority of daily Lean user activity, rather than every niche intent**. Our intent categories also closely mirror the structure of Zulip channels: for example, *lemma search* corresponds to #Is there code for X, *metaprogramming/tactic programming* aligns with #metaprogramming/tactics, and categories such as *proof engineering* and *type-class/instance* naturally arise from discussions in #lean4 and #mathlib4.
>
> Meanwhile, **our intent discovery pipeline is fully generalizable**. The framework is **designed to be scalable and can be applied to any additional corpus** of discussions to uncover new user intents. We plan to utilize this extensibility to broaden the coverage of Lean Finder in future work.
>
> > **Q6.** Small-scale user study
>
> 1. The user study **requires participants to have both Lean and mathematics expertise**, which makes **recruiting a larger pool difficult** given the small size of this expert community. This is also reflected by the low engagement in “job posting” threads on Lean Zulip for positions requiring Lean + math expertise.
>
> 2. Despite having 5 participants, the study includes 128 query–responses comparisons, **yielding a large amount of evaluative evidence**. The results show a **substantial advantage for Lean Finder**: its count of being selected as most helpful is almost **double** that of the strongest baseline.
>
> > **Q7.** How do you make Lean Finder compatible with multiple versions of Lean/Mathlib?
>
> Thanks for the suggestions! It is indeed important to keep our tool version-aware for fast-growing Mathlib and Lean. We promise update our web service to handle different versions of Mathlib, and let users pick the version they want to use.
>
> > **Q8.** Regarding the 5 user intent categories, did you test whether they can cover the intents of newly collected user queries (beyond the initial 693)?
>
> The user queries collected from our website for DPO fine-tuning beyond the initial 693 queries are all successfully clustered in the 5 intent categories using LLM, and confirmed by human inspection. **This shows that our 5 intent clusters generalize to new real user queries**. We show the clustering results of the additional unique queries used in DPO fine-tuning:
>
> | User intent | Count |
> |:----------------:|:---------:|
> |Searching for existing code/lemmas |136 |
> |Metaprogramming / tactic programming |66 |
> |Type‑class,  instance,  axiom control |35 |
> |Proof engineering & everyday Lean usage |169 |
> |Library design & large‑scale formalization |44 |
>
> The distribution of query types on our website shows a similar pattern to that of our dataset. For example, lemma search and proof-engineering queries each make up roughly 41% of our dataset, and both categories also exceed 30% in the clustering results above.

---

> ### Author Response · Authors · 2025-11-26
> **Anticipating Further Discussions**
>
> Dear Reviewer xHpC:
>
> We would like to kindly remind you that the author-reviewer discussion period has started for several days now and will end soon.
>
> We would greatly appreciate it if you could review our responses to your initial comments at your earliest convenience.
>
> This will enable us to address any additional queries or feedback you might have before the discussion period ends.
>
> Should our responses sufficiently address your concerns, we respectfully request that you consider raising the rating of our work.
>
> Thank you very much for your attention, time, and efforts!
>
> Best regards,
>
> Authors of Submission #12884

---

### Author Response · Authors · 2025-11-20

We deeply appreciate the feedback and suggestions from all four reviewers. We are pleased that **all four reviewers recognized that our work targets a core pain point in Lean theorem proving and offers significant community value for the Lean and formal mathematics ecosystem.** We thank all four reviewers for **acknowledging our main contribution, which is to build a strong and user-centered semantic search engine for Lean and Mathlib.** We also thank reviewers xHpC, jNAc, and XmGe for emphasizing our novel, user-intent-driven data synthesis pipeline that addresses the scarcity of real user data for Lean, as well as our rigorous training designs.

We would like to address a common concern, which is the lack of comparison of Lean Finder with the newest tool in semantic search for Lean, called Lean Explore. After ensuring the test queries’ ground truth Lean code appear in Lean Explore’s static database, we show results comparing Lean Finder and Lean Explore on all input types that Lean Explore supports:

| Model                          | Input Type                      | R@1  | R@5  | R@10 | MRR  |
|:-------------------------------|:-----------------------------|:----:|:----:|:----:|:----:|
| Lean Finder (ours)            | Informalized statement       | **65.7** | **89.0** | **93.2** | **0.76** |
| Lean Explore                  | Informalized statement       | 35.0 | 61.8 | 68.0 | 0.46 |
| |                   |      |      |      |      |
| Lean Finder (ours)            | Synthetic user query         | **57.3** | **85.4** | **91.2** | **0.69** |
| Lean Explore                  | Synthetic user query         | 26.3 | 51.3 | 60.1 | 0.37 |
| |                              |      |      |      |      |
| Lean Finder (ours)            | Augmented statement          | **86.8** | **96.7** | **98.0** | **0.91** |
| Lean Explore                  | Augmented statement          | 85.8 | 93.6 | 95.2 | 0.89 |

Lean Finder still outperforms Lean Explore, especially on natural language queries like informalized statements and user queries. This shows Lean Finder’s strengths and advantages still hold when compared with the newest tool like Lean Explore.

We address other questions and concerns in individual responses.

---

### Author Response · Authors · 2025-11-30
**Rebuttal Summary**

Dear PC, AC, and reviewers:

Since further public discussions are no longer allowed, we would like to post a rebuttal summary of our comments, justifications, and updates made during this rebuttal period. We sincerely appreciate all the reviewers for their thoughtful suggestions that help improve the paper quality.

We appreciate the **unanimous recognition from all reviewers** that our work **targets a core challenge in Lean theorem proving and offers high community value**. We are especially grateful to reviewers xHpC, jNAc, and XmGe for highlighting our **rigorous training designs and novel data synthesis pipeline**. This approach, as recognized by the reviewers, **successfully addresses the scarcity of real-world data** and **bridges the critical gap between real user queries and the mechanical informalization of Lean code, which is overlooked by previous works**.

Notably, we have comprehensively **conducted a significant amount of experiments**, and have **addressed all issues from reviewers**. Specifically, during the rebuttal, reviewer 2WYg expressed the **willingness of significantly increasing our score**, as stated in [the discussion](https://openreview.net/forum?id=5XNnnbEcu5&noteId=blfcnr9KcZ).

Below, we list the main questions raised by the reviewers, and explain how we comprehensively addressed all questions.

> **Lean Finder consistently outperforms all baselines**

1) Additional experiments compare our Lean Finder with the newest search engine Lean Explore, Lean Search with augmentation mode, and one of the state-of-the-art embedding model Qwen3 Embedding 8B. Results ([Lean Explore comparison](https://openreview.net/forum?id=5XNnnbEcu5&noteId=jdfe36Pdoo),  [Lean Search with augmentation mode comparison](https://openreview.net/forum?id=5XNnnbEcu5&noteId=GXotJfGgkB), [Qwen3 Embedding 8B comparison](https://openreview.net/forum?id=5XNnnbEcu5&noteId=DoiARvQNHg)) show that **Lean Finder still significantly outperforms all the additional baselines on all input types, achieving at least 11.3% relative improvement on recall@1**.

2) We integrated an additional search engine that supports proof state queries called Lean State Search with the reference LLM theorem provers following the same RAG settings as we integrated Lean Finder. The [results](https://openreview.net/forum?id=5XNnnbEcu5&noteId=8Ijc8zLXgO) show that **integration with Lean Finder yields more consistent and larger gains in theorem proving performance**.

3) Furthermore, the results (Table 3) show that our DPO fine-tuned model that leverages user feedback is the **most preferred in the user study**, achieving nearly double the "most helpful" votes compared to the strongest baseline. This demonstrates **Lean Finder's real-world value**, and as noted by **Reviewer 2WYg** during the [discussion](https://openreview.net/forum?id=5XNnnbEcu5&noteId=NLWrhdMYnV), these **user study results are very convincing**.


> **Lean Finder also outperforms on short, key word based queries**

Following Reviewer 2WYg's suggestion to address extremely short user queries, we evaluated Lean Finder on the dataset provided by Lean Explore, which consists exclusively of keyword-based inputs, matching the described query type by reviewer 2WYg. **[Results](https://openreview.net/forum?id=5XNnnbEcu5&noteId=R1P6AaUAmg) show that Lean Finder significantly outperforms both Lean Explore and Lean Search on this distinct query style, demonstrating our system's robustness across diverse user behaviors**.

> **Our synthetic user queries are of high quality and strongly align with real user data**

1) **Our high-quality synthetic user query largely improves Lean Finder’s performance**. We performed ablation studies to isolate the effect of including synthetic user queries in training our model (Appendix K). The results show that the model trained without them, while keeping training steps the same, is much less preferred by real users in the user study compared to the model trained with them. This **validates the high semantic quality of our synthetic data ([acknowledged by Reviewer 2WYg](https://openreview.net/forum?id=5XNnnbEcu5&noteId=NLWrhdMYnV)) and the strong alignment between our synthetic queries and real user queries**.

2) This high quality is further demonstrated by Figure 2, where our synthetic user query embeddings align with real user queries much better than informalization of Lean 4 code that the previous search engine relies on.

---

> ### Author Response · Authors · 2025-11-30
> **Rebuttal Summary Part 2**
>
> > **For additional explanations or justifications, we direct to the rebuttal and discussions with reviewers**:
>
> Reviewer xHpC:
> 1. Addressing concern on web scraping ([Q1](https://openreview.net/forum?id=5XNnnbEcu5&noteId=u3w702qYbZ))
> 2. Validation measures in data synthesis process ([Q2](https://openreview.net/forum?id=5XNnnbEcu5&noteId=u3w702qYbZ)),
> 3. Coverage of user intents ([Q5](https://openreview.net/forum?id=5XNnnbEcu5&noteId=OAU7wSCkhS))
> 4. Addressing user study scale ([Q6](https://openreview.net/forum?id=5XNnnbEcu5&noteId=OAU7wSCkhS))
>
> Reviewer jNAc:
> 1. Further analysis on synthetic data quality and diversity ([Q1](https://openreview.net/forum?id=5XNnnbEcu5&noteId=DoiARvQNHg))
>
> Reviewer 2WYg:
> 1. Case study on performance difference between Lean Finder and Lean Search with augmentation mode ([Q2](https://openreview.net/forum?id=5XNnnbEcu5&noteId=GXotJfGgkB))
> 2. Justification for using GPT-4o as one of the many baselines ([Q3](https://openreview.net/forum?id=5XNnnbEcu5&noteId=FVzUA2B01c))
> 3. Addressing concern on limited innovation in model architecture ([Q5](https://openreview.net/forum?id=5XNnnbEcu5&noteId=FVzUA2B01c))
> 4. Further addressing concern on potential mismatch of synthetic queries and real queries ([discussions part1](https://openreview.net/forum?id=5XNnnbEcu5&noteId=tG4E2Ov3Ib), [discussion part2](https://openreview.net/forum?id=5XNnnbEcu5&noteId=R1P6AaUAmg))
>
> Reviewer XmGe:
> The question about paper content was addressed in previous part
> ***
> We have comprehensively addressed all questions and concerns raised by reviewers. In general, Lean Finder **solves a core, overlooked bottleneck in current search engines by natively aligning them with human intent**. Our novel data synthesis pipeline and use of user feedback in preference alignment close the domain gap between real queries and mechanical code informalization, **addressing an unexplored blind spot in the existing Lean search literature**. **Our superior performance against multiple baselines and in real-world user studies demonstrates that Lean Finder offers significant value to the community**.

---

### Meta-Review · Area_Chair_QXVt · 2026-01-06

**Summary:**

This paper introduces a user-centered semantic search Lean Finder, tailored to the needs of mathematicians by analyzing and clustering the semantics of public Lean discussions, then fine-tuning text embeddings on synthesized queries that emulate user intents. Lean Finder is then aligned with mathematicians' preferences with diverse feedback signals. Experiments include evaluation of retrieval performance, a user study on real queries, and exploration of Lean Finder as a retrieval backbone for LLM-based provers, and show significance.

**Reviewer Concerns:**

Addressed main concerns:
* Incomplete baseline comparison (Reviewer xHpC, jNAc, 2WYg): Additional comparison with Lean Explore, Lean Search (augmentation mode), Qwen3-Embedding-8B, and provers with Lean State Search.
* Limited Methodological Innovation (2WYg): Strong enough justification of the contributions.
* Potential mismatch in length and style between synthetic queries and real user queries, and the use of a single query (2WYg): Provided detailed explanations.

Partially addressed concerns:
* Lack of Solutions for Lean Version Differences (Reviewer xHpC)
* Fine-tuning experiments are conducted solely on the DeepSeek-Prover-V1.5-RL 7B model (Reviewer jNAc)

**Reviewer Scores:**

Given that most of the core concerns are addressed by the authors' rebuttal with additional experimental results and detailed explanations, the reviewers are highly likely to increase their scores.

---

### Decision · Program_Chairs · 2026-01-26

Accept (Poster)